# ColdZyme® reduces viral load and upper respiratory tract infection duration and protects airway epithelia from infection with human rhinoviruses

Glen Davison[1] , Marlene Schoeman[1], Corinna Chidley[2], Deborah K. Dulson[3], Paul Schweighofer[4], Christina Witting[4], Wilfried Posch[4], Guilherme G. Matta[1], Claudia Consoli[5], Kyle Farley[2], Conor McCullough[2] and Doris Wilflingseder[4,6]

[1] *School of Natural Sciences, University of Kent, Kent, UK*

[2] *School of Sport and Exercise Science, University of Derby, Derby, UK*

[3] *School of Biomedical, Nutritional and Sport Sciences, Faculty of Medical Sciences, Newcastle University, Newcastle, UK*

[4] *Institute of Hygiene and Medical Microbiology, Medical University of Innsbruck, Innsbruck, Austria*

[5] *College of Biomedical and Life Sciences, Cardiff University, Cardiff, UK*

[6] *Infectiology and Virology Unit, University of Veterinary Medicine Vienna, Vienna, Austria*

Handling Editors: Harold Schultz & Mike Stembridge

The peer review history is available in the Supporting Information section of this article (https://doi.org/10.1113/JP288136#support-information-section).

**Abstract figure legend** These experiments provide robust evidence of the physiological mechanisms that explain the effects of ColdZyme, including protection against upper respiratory tract infection (URTI) and viral release from infected airway epithelial cells and better maintained cell integrity. *In vivo*, the duration for which URTI symptoms were experienced by participants was lower with ColdZyme treatment, which also resulted in fewer absence days. Oropharyngeal swabs collected over the first 7 days post-symptom onset showed lower rhinovirus viral load with ColdZyme treatment compared with placebo. *In vitro*, human airway epithelial cells were protected from infection with rhinovirus by the application of ColdZyme. Treatment also reduced virus release, both apically and basolaterally. As a result of lower viral infectivity, epithelial cells suffered less damage and inflammation, which further explains the *in vivo* findings of lower symptom scores, shorter episode duration and fewer absence days with ColdZyme treatment.

**Abstract** Upper respiratory tract infection (URTI) has a significant economic and social impact and is a major factor compromising athletes' training and competition. The effects of ColdZyme® Mouth Spray on URTI were investigated using an *in vivo* study in athletes, combined with a novel *in vitro* air–liquid interface human airway model. Endurance athletes were randomised to ColdZyme ($n = 78$) or placebo ($n = 76$) and monitored over 3 months. They completed daily symptom and training logs and collected throat swabs over 7 days during perceived URTI. *In vitro* studies examined rhinovirus infectivity and epithelial barrier integrity of airway epithelial cells. Eighty-two *in vivo* episodes were analysed with significantly lower ($P = 0.012$) episode duration in the ColdZyme *vs.* Placebo group (mean ± SD, 6.2 ± 2.6, (median [interquartile range]) 5.5 [4–9] days *vs.* 10.7 ± 10.2, 7.0 [5–11]). There was no difference in incidence ($P = 0.149$). Training absence and symptom ratings were lower ($P < 0.05$) in the ColdZyme group. Swabs were returned for 50 episodes, with at least one pathogen detected in all (rhinovirus was most abundant). Absolute quantification (qPCR) for rhinovirus revealed a significantly lower 7-day area under the curve in ColdZyme *vs.* placebo (median reduction, 94%, $P = 0.029$). *In vitro*, viral load was significantly lower (median reductions 80–100%), and epithelial barrier integrity better maintained, and no virus was detected by immunofluorescence analyses of pseudostratified epithelia, with ColdZyme treatment (all $P < 0.05$). ColdZyme is beneficial for reducing URTI duration, symptom ratings and missed training days. These novel data suggest that the mechanisms involve the protection of epithelial cells against rhinovirus infection and damage.

(Received 18 November 2024; accepted after revision 7 February 2025; first published online 27 February 2025)

**Corresponding author** G. Davison: School of Natural Sciences, University of Kent, Canterbury, CT2 7PE, UK. Email: g.davison@kent.ac.uk

**Key points**

- Upper respiratory tract infections (URTI) are a common complaint in the general population and athletes alike, with social, well-being and economic consequences, including performance detriments in athletes and reduced work productivity in the general population.
- Strategies to minimise the risk of contracting a URTI and/or reduce the time taken to clear an infection are desirable to athletes and the general population alike.
- The present study employed an *in vivo* study with athletes in combination with a novel *in vitro* human airway cell model to examine the effects of ColdZyme Mouth Spray on URTI and viral infectivity.
- The duration for which URTI symptoms persisted was lower with ColdZyme treatment, which also resulted in fewer training absence days.
- Swabs from participants in the *in vivo* study and supernatants from the *in vitro* studies showed lower rhinovirus viral load with ColdZyme treatment compared with placebo or control.

**Glen Davison** (PhD, Loughborough) is a Professor of Sport & Exercise Sciences, specialising in Exercise Immunology. His research interests focus on *in vivo* assessment of epithelial barrier integrity; how the human immune system responds to endurance exercise; and upper respiratory tract infection risk in athletes. Prof. Davison holds British Association of Sport & Exercise Sciences (BASES) accreditation and Chartered Scientist (CSci) status. **Doris Wilflingseder** studied Zoology at the University of Innsbruck, specialising in cell biology, immunology and disease modelling. After a visiting scientist stay at University College London, and an academic career at the Medical University of Innsbruck, she was appointed University Professor of Infectiology at the Ignaz Semmelweis Institute and the Veterinary University of Vienna. Dr Wilflingseder has received several awards, including the Austrian State Award 2021 for promoting Alternatives to Animal Testing, and the Austrian Microbiology Prize.

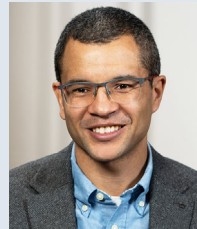
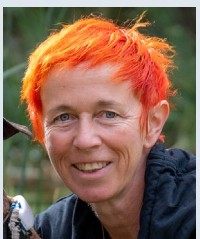

## Introduction

Upper respiratory tract infections (URTI) can have significant economic and social impacts (e.g. absence from work, healthcare costs, increased morbidity, reduced feelings of well-being, health and quality of life and reduced social interaction), and are one of the most common reasons a patient may consult a physician (Bramley et al., 2002; Hashem & Hall, 2003; Hellgren et al., 2010). They are also among the most common types of illness in athletic populations (Cannon, 1993; Gleeson & Walsh, 2012; Schwellnus et al., 2021), second only to musculoskeletal injury as a reason for athletes to seek medical attention at both the summer and winter Olympic Games (e.g. Engebretsen et al., 2010, 2013; Robinson 2002; Valtonen et al., 2019). Such infections can result in a loss of training days, and/or directly compromise training or competition performance (Cunniffe et al., 2011; Keaney et al., 2019; Nieman, 1994; Nieman & Wentz, 2019; Pyne et al., 2005; Reid et al., 2004; Svendsen et al., 2016; Walsh et al. 2011). Many of the general risk factors for respiratory infections are largely the same for athletes and non-athletes. However, in athletes, illnesses also seem to cluster more frequently around periods of intensive training and/or competition (Hellard et al., 2015; Svendsen et al., 2016), so strategies to reduce the risk of contracting these illnesses, or shorten recovery time, will be of direct benefit to them. For example, previous research has reported that World Championship and Olympic medal-winning athletes reported fewer URTI symptoms leading up to competition, compared with those that did not win medals (Raysmith & Drew, 2016). Effective interventions may also limit the risk of spreading infection to other athletes, but also support teams, family members, colleagues and vulnerable individuals, which also aligns with previous observations showing associations between URTI risk and household illness in elite athletes (Keaney et al., 2021; Keaney et al., 2022). Beyond the benefits to athletes, reducing the risk of spreading URTIs may also have a positive effect on the economic and social impacts associated with URTIs.

To date, most strategies to minimise the risk of contracting a URTI and/or reduce time taken to clear an infection have focused on avoidance of exposure and minimising the controllable risk factors that are associated with lowered immune defence (e.g. intensified physical training, life stressors). However, these factors may be unavoidable, or at least difficult to avoid for many athletes (Keaney et al., 2019; Svendsen et al., 2016) or other groups (e.g. those in demanding (physical and/or psychological) occupations), as well as the wider general population. An alternative strategy that has received little attention, is the use of products that might inhibit viral infectivity by limiting either viral entry or replication/propagation of the virus after initial exposure. Most URTIs are caused by viral infection, with over 200 known viruses – the most common being rhinoviruses, coronaviruses, influenza viruses, adenoviruses, parainfluenza viruses, respiratory syncytial viruses and enteroviruses (Heikkinen & Järvinen, 2003). Infection is initially established in the mucosa of the nasopharynx before spreading to other regions (Winther et al., 1986), with local symptoms typically beginning in the throat before nasal congestion, rhinorrhoea, sneezing and cough tend to develop (Witek et al., 2015). Receptors for virus–cell adhesion (e.g. ICAM-1 with rhinovirus) may be more abundant on the epithelial cells in the airways (e.g. Jacobs et al., 2013), which can also explain the pattern of symptom progression within the upper respiratory tract.

It is possible, therefore, that inhibiting viral propagation in the pharynx area may prevent or shorten the duration of a viral URTI. ColdZyme Mouth Spray (ColdZyme) consists of a hyperosmotic glycerol solution containing cold-adapted trypsin from the Atlantic cod (*Gadus morhua*) that is orally sprayed onto the pharynx, creating a temporary barrier. It has been shown to reduce URTI duration in a number of studies in healthy and clinical populations (e.g. Clarsund & Bråkenhielm Persson, 2016; Clarsund et al., 2017; Clarsund, 2017), and we have previously shown a reduction in URTI symptom duration in an open-label trial with ColdZyme in endurance athletes (Davison et al., 2021). To be effective, the spray must be applied regularly, with the manufacturer's instructions for use (IFU) being one application every 2 h, up to a maximum of six times per day. Indeed, in our previous study (Davison et al., 2021), participants with better compliance with the IFU had better outcomes than those with poorer compliance (i.e. ≤4 applications per day). *In vitro*, Stefansson et al. (2017) reported broad antiviral activity of ColdZyme resulting in the deactivation of 64–100% of virus activity for common URTI-causing pathogens (influenza virus, rhinovirus, adenovirus and coronavirus). It has also been shown to prevent viral infectivity (i.e. blocked cell binding, uptake and infection) and to maintain epithelial integrity using *in vitro* human respiratory air–liquid interface and organoid models (Posch et al., 2021 (SARS-CoV-2); Zaderer et al., 2022 (SARS-CoV-2 BA.4/5 variants)). *In vivo*, ColdZyme treatment has been shown to reduce the duration of the common cold by 54% in healthy adults inoculated with rhinovirus-16 (Clarsund et al., 2017). In a clinical case study with an immunodeficient 12-year-old boy, Clarsund et al. (2016) found that treatment with ColdZyme reduced reported URTI symptoms and resulted in a threefold reduction in missed school days due to illness. Studies thus far have shown promising results for ColdZyme; however, no placebo-controlled trials have examined whether such products can reduce URTI incidence, symptoms and episode duration in athletic populations.

The aims of this study were, therefore, to assess the efficacy of ColdZyme on URTI duration, incidence, symptom ratings and missed (or reduced) training in competitive endurance athletes under free-living conditions, in a prospective, randomised, placebo-controlled trial. We provide novel insights into the mechanisms of action with the inclusion of swabs for pathogen detection and viral load monitoring. To gain a greater understanding of the mechanisms for how any protective effects are elicited, we also performed *in vitro* studies examining viral infectivity and epithelial barrier resistance in a cell culture tissue model (using normal human airway epithelial cells) incubated with human rhinovirus in the presence or absence of ColdZyme.

## Methods

### *In vivo* study

**Ethical approval.** The study was conducted in accordance with the *Declaration of Helsinki* and approved by the ethics committees of the authors' universities (SSES REAG, University of Kent, ref: 20_20_21). All participants were informed, both verbally and in writing, of the nature and risks of the study before giving their written consent to take part. The trial was prospectively registered on the ISRCTN registry (ISRCTN18133939, https://doi.org/10.1186/ISRCTN18133939).

**Type of study.** The study was a prospective, double-blind, parallel group, randomised controlled trial.

Power calculation: the data from Davison et al. (2021) were used to estimate an effect size of ∼0.6 (for the primary outcome, episode duration) for differences between placebo and ColdZyme, which would require a total of $n = 72$ compliant URTI episodes to provide 80% power at 5% alpha. To protect against drop-out, estimated from our previous study, (+5% = 4 additional episodes) and poor IFU compliance (∼15% = 12 additional episodes), we aimed to recruit sufficient participants to produce ∼88 URTI episodes in total. Based on expected incidence rates, we estimated that around 114 participants would be required to achieve this. However, due to a lower-than-expected overall incidence rate, more participants (∼150) were ultimately required to attain the desired number of URTI episodes.

**Participants.** Endurance-trained athletes (e.g. long-distance runners, triathletes and cyclists) were recruited. Participants were excluded if on long-term medication; currently smoking; allergic to any of the ingredients in ColdZyme; had any other current medical conditions that might be aggravated by the use of the product; were currently using any medication (except for contraceptives), or food supplements; were currently using any other relevant products or supplements (nutritional or otherwise) that might influence the common cold; were currently taking part in another study that might compromise the results of this study; or were pregnant, breastfeeding or planning to become pregnant during the study.

**Design.** Athletes were monitored over a 3-month period of using the ColdZyme (or placebo). During this period, they completed self-report training logs, and the Jackson common cold questionnaire (Jackson questionnaire, Jackson et al., 1958) using an online platform (Qualtrics, Provo, UT). Participants were sent a personalised link by email (automatically sent by the software) each day at 20:00 for completion. If not completed, a reminder was sent 36 h later (08:00). No further reminders were sent automatically by the system. The online logs were also used for participants to record their daily product usage. The study period was in the UK winter months. Tranche 1 took place between November 2022 and May 2023, and Tranche 2 took place between October 2023 and May 2024. Tranche 1 took place in southeast England (East Kent and Medway areas) and the East Midlands. Tranche 2 took place in southeast England (including London), the East Midlands and northeast England (Newcastle-upon-Tyne).

Participants were randomised to either placebo or treatment group using an online platform (www.randomization.com, plan generated 06/10/2021) using block-randomisation with blocks of eight (i.e. every block of eight would have equal numbers of placebo and treatment, but it was completely randomised within each block). Randomisation was also allocated at each site independently so that groups were matched within each location (i.e. complete blocks were allocated to each site). The allocation schedule was concealed from investigators involved in recruiting the participants.

**Treatment.** The product and placebo were manufactured at Enzymatica's contract manufacturer Recipharm Parets (Spain), which is good manufacturing practice certified (ISO 13485). Placebo was identical in appearance and taste (flavoured with menthol in the same way as the ColdZyme product). The manufacturing site was provided with the randomisation plan so they could label treatment or placebo products accordingly, keeping the study fully double-blind. The randomisation plan was sent electronically in an encrypted and password-protected file, which was managed by a technician from the University of Kent not otherwise involved in the study. The unlock code was only known to the technician and the contract manufacturer and was not revealed until completion of all analyses.

Participants were provided with 6 × 20 ml bottles each and asked to use sprays in accordance with the manufacturer's instructions for treatment of a suspected common cold/URTI (at the first self-perceived signs of URTI). Briefly, this included instructions to spray two times (one dose) every second hour up to six times daily. In addition to using the spray at 'first signs' of a URTI, participants were instructed to optionally use sprays preventatively at times when they felt this necessary (i.e. increased exposure risk), although this was not compulsory. They were also required to record the number of uses each day, using the daily online training and illness logs, which allowed for compliance with IFU to be established and to determine which participants (if any) did use it preventatively.

Participants were not restricted from using over-the-counter (OTC) medication if they felt it necessary, but were required to record any usage in their illness log.

Participants were asked to complete a daily training log to calculate training load (using the session rating of perceived exertion (RPE) method, Foster et al., 2001) so that training and URTI symptoms could be quantified as described in our previous work (e.g. Davison et al., 2021; Jones et al., 2014). They were required to indicate if they were suffering from any of the illness symptoms listed on the Jackson questionnaire, giving a numerical rating for the severity of any reported symptoms (0 not present, and 1, 2, 3 for mild, moderate or severe, respectively). Participants were provided with self-swab kits, which contained 1 ml stabilisation buffer (DNA Genotek, Stittsville, Canada) and instructed in their use, which has been shown to be effective in other surveillance studies with self-swab methods (e.g. Goff et al., 2015). They were required to take a swab if they perceived experiencing a URTI episode, on days 1, 3, 5 and 7 (where day 1 was the first day that symptoms presented).

An episode of URTI was defined using the Jackson criteria: scores for each of eight symptoms were summed for each day to generate a total Jackson score, and an episode was defined as those lasting ≥3 days and with either (i) a total Jackson symptom score of ≥6 + subjective impression of having a cold (question 1), or (ii) a total Jackson symptom score of ≥6 + nasal discharge for at least 3 days, or (iii) a total Jackson symptom score <6 + subjective impression of having a cold + nasal discharge for at least 3 days. We initially used the daily threshold of 14 (as applied by Martineau et al. 2015), but the Cohen et al. (2003) method was chosen to better align with the original Jackson scoring criteria. However, there was very little difference in overall outcome regardless of which method was applied, with the only difference being that an additional case was included if using the Cohen et al. (2003) method ($n = 82$) compared with the Martineau et al. (2015) method ($n = 81$). Data reported in this paper are based on the Cohen et al. (2003) daily 'cut-off' value of ≥6).

**Analytical methods.** All nucleic acid extraction, pre-amplification, reverse transcription and qPCR analysis were performed in an ISO 9001:2015 and GCLP-accredited facility (Central Biotechnology Services, Cardiff University, UK). Nucleic acids were extracted from the samples (400 µl from swab stabilisation solution) using MagMAX Viral/Pathogen Ultra Nucleic acid extraction kits (ThermoFisher) and PCR was performed with the TrueMark Respiratory Panel 2.0 array cards (ThermoFisher) on a QuantStudio 12K Flex Real-Time PCR System (ThermoFisher). Pathogens were normalised against RNase P (Guest et al., 2020; Miranda et al., 2021), using the Delta-Delta Ct method (to give relative quantification: Rq). Results are expressed as area under the (Rq) curve (AUC) (days 1–7 inclusive). Following this initial screening, absolute quantification was performed on the two most abundant pathogens (*Haemophilus influenzae* and rhinovirus targets) with specific targeted qPCR assays performed in triplicate (with replicate values then averaged). Each target assay was duplexed with RNase P as an endogenous control for normalisation, to determine absolute quantification (copies) for these targets. For the absolute quantification, a standard curve was generated using a serial dilution of the assay amplification control sample (with known copy numbers per microlitre) to allow the calculation of copies per target. The standard curve was included in every plate together with negative controls. The PCR efficiency was close to 100% for each assay.

### *In vitro* human airway model

**Cell culture of tissue models and ColdZyme treatment.** Human airway epithelial (HAE) cells: normal human bronchial epithelial cells (NHBE, Lonza, cat# CC-2540S) are available in our laboratory and routinely cultured in air–liquid interface (ALI) as described in our previous work (e.g. Posch et al., 2021; Zaderer et al., 2022). Briefly, cells were cultured in a T75 flask for 2–4 days until they reached 80% confluence. The cells were treated using animal-free TrypLE (Gibco) and seeded onto GrowDexT (UPM)-coated 0.33 $cm^2$ porous (0.4 µm) polyester membrane inserts with a seeding density of $1 \times 10^5$ cells per Transwell (Costar, Corning, New York, NY, USA). The cells were grown to near confluence in a submerged culture for 2–3 days in a specific epithelial cell growth medium according to the manufacturer's instructions. Cultures were maintained in a humidified atmosphere with 5% $CO_2$ at 37°C and then transferred to ALI culture. The epithelium was expanded and differentiated using airway media from Stemcell. The number of days in

development was designated relative to the initiation of ALI culture, corresponding to day 0. One hub of ColdZyme mouthspray was applied to the apical side of the fully differentiated epithelia prior to infection using rhinoviruses (RVs) RV35 and RV48. This corresponded to approximately 50 μl of liquid, evenly dispersed over the tissue culture. The apical application was carefully performed so as not to mechanically disrupt the epithelial surface.

HeLa cells were used to expand human RV35 and RV48 (ATCC, ATCC-VR-508, ATCC-VR-1875). Viral loads were detected by a rhinovirus-specific real-time RT-PCR on several days post-infection (dpi).

**Transepithelial electrical resistance (TEER) measurement.** TEER values were measured using an EVOM volt-ohm-meter with STX-2 chopstick electrodes (World Precision Instruments, Stevenage, UK). Measurements on cells in ALI culture were taken immediately before the medium was exchanged. For measurements, 0.1 and 0.7 ml of medium were added to the apical and basolateral chambers, respectively. Cells were allowed to equilibrate before TEER was measured. TEER values reported were corrected for the resistance and surface area of the Transwell filters.

**Staining and high content screening.** To visualise rhinovirus infection in 3D tissue models, cells were infected with RV35 or 48 obtained from repositories (see below) and analysed for characteristic markers in infection experiments on day 3 post-infection. After rhinovirus exposure, 3D cell cultures were fixed with 4% paraformaldehyde. Intracellular staining was performed using 1× Intracellular Staining Permeabilization Wash Buffer (10X; BioLegend, San Diego, CA, USA). Cultures were stained using acetylated tubulin-488 for cilia, Hoechst 33 342 for nuclei, and MUC5AC-555 and phalloidin-555 for detecting mucus-producing cells and F-actin. Intracellular rhinovirus was detected using Alexa594-labelled dsRNA-Ab. The Alexa594-labelling kit was purchased from Abcam, Cambridge, UK (cat# ab269822). After staining, 3D cultures were mounted in Mowiol. To study these complex models using primary cells cultured in 3D and to generate detailed phenotypic fingerprints for deeper biological insights in a high throughput manner, the Operetta CLS System (PerkinElmer, Waltham, MA, USA) was applied. For 3D analyses, Harmony Software was used, and the experiments were repeated independently four times (twice using RV35, twice with RV48). For each condition, three independent experiments were performed (from distinct areas of the culture), each including 300–500 cells per sample per analysis.

**Viruses.** Rhinovirus 35 and 48 obtained from ATCC (RV35 cat# VR-1999TM; RV48 cat# VR-1875TM) were propagated in a susceptible HeLa cell line as described above, characterised by rhinovirus-specific real-time PCR and used subsequently to infect cells.

**Rhinovirus-specific real-time PCR.** Viral loads were analysed in HeLa cells and 3D ALI NHBE cultures on several dpi by using a rhinovirus-specific real-time PCR.

**Data analysis.** All *P*-values for comparisons between groups are presented as the one-sided value owing to the directional (i.e. one-tailed) hypotheses (with directional hypotheses pre-specified in the trial pre-registration). All data analysis was conducted utilising IBM SPSS statistics version 29 (IBM, Armonk, NY).

Data were checked for normal distribution prior to analysis using Z-scores for skewness and kurtosis. Data that did not have a normal distribution were normalised via a natural log or square root transformation prior to analysis. Data that could not be normalised by log or square root transformation were analysed with non-parametric tests. When summary data are presented as the mean, the SD is reported and when medians are presented, the interquartile range is shown.

The following transformations were required to allow parametric tests to be used:

Log transformation: URTI episode duration (whole group); total Jackson score (whole group), mean daily Jackson score (whole group); URTI episode duration (positive swab group); total Jackson score (positive swab group), peak daily Jackson score (positive swab group).

Square root: peak daily Jackson score (whole group); individual Jackson symptoms (whole group) – sneezing, headache, malaise, nasal discharge, nasal obstruction, sore throat; individual Jackson symptoms (positive swab group) – sneezing, headache, malaise, nasal discharge, nasal obstruction, sore throat; days training missed due to URTI (whole group and positive swab group); absolute quantification PCR data for rhinovirus.

The following variables could not be normalised so were analysed with non-parametric tests. The number of URTI episodes per person; individual Jackson symptoms (whole group and positive swab group scores) – chilliness, cough; training load data (percentage of healthy) during and 7 days after URTI episodes; days training affected by URTI (whole group and positive swab group); days training reduced due to URTI (whole group and positive swab group); Rq PCR data (7 days AUC) for *H. influenzae* and for rhinovirus, and absolute quantification data for *H. influenzae*.

Group comparisons were made using independent sample *t* tests (or for non-parametric statistics, the Mann–Whitney U test). For the *in vitro* infection levels

**Table 1. Upper respiratory tract infection (URTI) episode duration (primary outcome measure).** *n* = 82 episodes

| Variable | Placebo | ColdZyme | *P* |
|---|---|---|---|
| **URTI episode duration (days)** | | | |
| Mean ± SD | 10.7 ± 10.2 | 6.2 ± 2.6 | 0.012 |
| Median (interquartile range) | 7.0 (5.0–11.3) | 5.5 (4.0–8.8) | |

Statistical tests: independent *t* test on transformed (normalised) data.

and TEER values, with $n \leq 5$ samples (biological replicates), and normal distribution could not be confirmed, non-parametric tests were utilised: the Kruskal–Wallis test with Bonferroni-corrected Mann–Whitney U *post hoc* tests used where required to compare conditions.

## Results

The first subject was enrolled in the study on 11 November 2022 and the final subject completed on 19 May 2024. A total of 164 subjects were enrolled, but 10 were lost to follow-up ($n = 4$ placebo; $n = 6$ ColdZyme, see Fig. 2). Analysis was completed on $n = 154$ (placebo $n = 76$, age at enrolment 34 ± 11 years (54 male, 22 female); ColdZyme $n = 78$, age at enrolment 36 ± 10 years (48 male, 30 female)). Athletes were all endurance athletes in current training, ranging from club-level athletes to international-level athletes (self-reported into the following categories: highly competitive/elite (e.g. regional, national, international), $n = 14$; highly competitive sub-elite (e.g. club, university), $n = 52$; lower-level competitive, $n = 47$; or not disclosed, $n = 41$). At least one URTI episode was recorded during the study period in 45% of all participants (45% placebo, 45% treatment). The remaining 55% either did not report any subjective episodes, or reported symptoms that did not attain the criteria to be counted as an episode (i.e. duration ≤2 days or did not meet the Jackson scoring criteria).

In total, 91 episodes were recorded by all participants, but nine episodes (~10%) were excluded for poor compliance with IFU. Poor compliance was determined as using the spray less than 80% of the recommended six times per day, which equates to using it less than five times per day). Participants who were included also had at least 90% compliance with daily log completions. Episode duration was shorter in the ColdZyme group compared to placebo group (see Table 1). There was no difference between groups in the average incidence rate (episodes per person, $P = 0.149$, see Table 2).

### Pathogen screening

At least one positive pathogen was detected in all swabs. In total, 24 different pathogens were detected across samples (adenovirus, *H. influenzae*, rhinovirus, Epstein-Barr virus, human herpesvirus 6, influenza, parainfluenza, respiratory syncytial virus, metapneumovirus, measles, mumps, coronaviruses, MERS CoV, SARS-CoV, SARS-CoV2 (Covid-19), enterovirus, *Bordetella*, *Bordetella pertussis*, *Klebsiella pneumoniae*, *Moraxella catarrhalis*, *Mycoplasma pneumoniae*, *Staphylococcus aureus*, *Streptococcus pneumoniae* and *Pneumocystis jirovecii*), many of which were co-infections. Only *H. influenzae* (30 episodes: 14 placebo, 16 ColdZyme) and rhinovirus (50 episodes: 24 Placebo, 26 ColdZyme) were detected with sufficient frequency in each group to permit statistical comparisons on relative viral and bacterial loads (see Table 3 and Fig. 3). Thirty-five of the participants also returned a healthy/asymptomatic swab sample, which was collected at a point when they had no perceived URTI and were generally well. Of these, 46% ($n = 16$) were negative for all pathogens on the panel. The other 54% ($n = 19$) returned a positive result for at least one pathogen (range 1–2 pathogens detected per person). These comprised adenovirus (one positive), *H. influenzae* (four positives), rhinovirus (14 positives), influenza (two positives), Epstein-Barr virus (one positive), human herpesvirus 6 (one positive), respiratory syncytial virus (one positive) and *S. pneumoniae* (two positives).

### Absolute quantification for *H. influenzae* and rhinovirus

Note: due to the high number of sub-types for rhinovirus, there are two separate assay sequences used in the Respiratory Pathogen array panel (named RV 1 of 2, ID: Vi99990016_po and RV 2 of 2, ID: Vi99990017_po, https://www.thermofisher.com/microbe-detection/taqman/query/), which is required to give full strain coverage, but results in some overlap. For this reason, both of these rhinovirus assays (Vi99990016 and Vi99990017) were performed with the results combined (using the highest value) for final results analysis.

### Daily training logs

Training 'load' was quantified as the product of session RPE and duration for each session (and summed each week). Data from 10 participants were excluded from this

**Table 2. Upper respiratory tract infection episodes and parameters recorded using the Jackson questionnaire (*n* = 82)**

| | | Placebo | ColdZyme | *P* |
|---|---|---|---|---|
| **Number of episodes per person** | | | | |
| Mean ± SD | | 0.45 ± 0.60 | 0.62 ± 0.79 | 0.149 |
| Median (interquartile range) | | 0 (0–1) | 0 (0–1) | |
| **Jackson score (whole episode)** | | | | |
| Mean ± SD | | 57.3 ± 52.4 | 37.8 ± 27.8 | 0.032 |
| Median (interquartile range) | | 37.0 (22.8–60.3) | 28.0 (18.0–53.5) | |
| **Jackson score (mean per day)** | | | | |
| Mean ± SD | | 5.7 ± 2.4 | 5.7 ± 2.4 | 0.438 |
| Median (interquartile range) | | 5.4 (4.1–7.0) | 5.7 (3.9–7.2) | |
| **Jackson score (episode peak)** | | | | |
| Mean ± SD | | 10.1 ± 4.0 | 9.2 ± 4.2 | 0.177 |
| Median (interquartile range) | | 9.5 (7.0–13.3) | 9.0 (6.0–11.0) | |
| **Jackson individual symptoms** | | | | |
| Sneezing | Mean ± SD | 6.1 ± 4.9 | 4.1 ± 3.3 | 0.026 |
| | Median (interquartile range) | 4.5 (2.8–9.0) | 4.0 (1.0–6.0) | |
| Headache | Mean ± SD | 3.9 ± 5.1 | 3.9 ± 4.5 | 0.478 |
| | Median (interquartile range) | 2.5 (0.0–5.0) | 3.0 (0.0–7.8) | |
| Malaise | Mean ± SD | 8.8 ± 10.1 | 4.5 ± 5.1 | 0.013 |
| | Median (interquartile range) | 5.5 (1.8–13.0) | 3.0 (0.0–8.5) | |
| Nasal discharge | Mean ± SD | 10.4 ± 7.9 | 6.8 ± 4.1 | 0.014 |
| | Median (interquartile range) | 8.5 (4.8–14.0) | 6.0 (4.0–9.0) | |
| Nasal obstruction | Mean ± SD | 10.3 ± 12.7 | 5.4 ± 5.1 | 0.028 |
| | Median (interquartile range) | 5.5 (2.8–10.8) | 4.5 (1.0–9.0) | |
| Sore throat | Mean ± SD | 6.3 ± 5.8 | 4.9 ± 3.8 | 0.102 |
| | Median (interquartile range) | 4.5 (2.0–9.0) | 4.0 (2.0–6.8) | |
| Chilliness | Mean ± SD | 2.3 ± 3.5 | 1.6 ± 3.3 | 0.158 |
| | Median (interquartile range) | 0.0 (0.0–3.5) | 0.0 (0.0–1.8) | |
| Cough | Mean ± SD | 9.1 ± 13.8 | 4.9 ± 5.5 | 0.067 |
| | Median (interquartile range) | 4.5 (2.0–8.3) | 3.0 (0.3–7.8) | |

Upper respiratory tract infection episodes recorded in 45% of participants (69/154): Placebo 45% (34/76), ColdZyme 45% (35/78). Statistical tests: independent *t* test on transformed (normalised) data for all variables except chilliness and cough (could not be normalised), which were compared with Mann–Whitney U tests.

analysis due to incomplete data recording by participants. For each participant, their average healthy ('normal') training load was calculated from all study weeks, excluding any weeks during which they reported a URTI episode and the 2 weeks before and after each episode. For the participants that recorded episodes, there was a significant decrease in training load during episodes in both the placebo group (mean 52.1% ± 60.6% of healthy load, median 38.1% [10.9–71.5%] of healthy load, *P* = 0.001) and ColdZyme group (mean 76.1% ± 62.5% of healthy load, median 69.6% [22.8–113.1%] of healthy load, *P* = 0.010), which returned to normal during the first week after symptoms subsided (*P* > 0.05 *vs.* healthy load).

### Treatment blinding checks

The placebo was flavoured the same as the ColdZyme product, and the external packaging and appearance were identical. To check success of the participant blinding, a separate online survey was sent to all participants after completion of the study. In total, 114 participants responded (the remaining 40 either did not respond or could not remember). The majority of the respondents reported that they did not know which group they were in (61%), with 22% guessing placebo and 18% guessing ColdZyme. These proportions were similar between the placebo and ColdZyme groups (respectively: did not

**Table 3. Viral/bacterial load by relative quantification (Rq) method from array card analysis (days 1–7 area under the curve (AUC))**

| Variable | Placebo | ColdZyme | P |
|---|:---:|:---:|:---:|
| *H. influenzae* bacterial load | *n* = 14 | *n* = 16 | 0.080 |
| ×10⁻³ Rq (*vs*. RNase P) AUC | 351 ± 669 | 37 ± 84 | |
| Mean ± SD | 74 (0–404) | 6 (0–19) | |
| Median (interquartile range) | | | |
| Rhinovirus viral load | *n* = 24 | *n* = 26 | 0.043 |
| ×10⁻³ Rq (*vs*. RNase P) AUC | 1427 ± 3141 | 337 ± 887 | |
| Mean ± SD | 129 (0–969) | 2 (0–93) | |
| Median (interquartile range) | | | |

Statistical tests: Mann–Whitney U tests.

know, 61% *vs*. 60%; guessed placebo, 23% *vs*. 21%; guessed ColdZyme, 16% *vs*. 19%), showing the broad success of the blinding.

### Use of OTC medication

Participants reported the use of OTC medication during URTI more frequently in the placebo group (65% of cases: 22/34 episodes) compared with the ColdZyme group (35% of cases: 17/48 episodes, $\chi^2$ *P* = 0.009). In the placebo group, the OTC medications used comprised analgesics only (e.g. paracetamol (acetaminophen), ibuprofen) in seven cases, cold and flu 'remedies' (e.g. Lemsip or similar; cough syrups or similar) in 14 cases and both treatments in one case. In the ColdZyme group OTC medication use comprised analgesics only in four cases, cold and flu remedies in 11 cases and both treatments in two cases.

### Adverse events reporting

No adverse events were reported by any participant.

### *In vitro* human airway model

**ColdZyme mouth spray protects from RV35 and RV48 infection in primary NHBE cells.** We monitored primary NHBE cell infection using RV35 and RV48 in the absence and presence of the ColdZyme mouth spray. A multiplicity of infection (MOI) of 0.01 was used to infect cells. One hub of ColdZyme was carefully sprayed from about 2.5 to 3 cm distance onto the apical side of fully differentiated, pseudostratified epithelia cultured at an ALI to realistically mimic the distribution within the oral cavity. ColdZyme was pre-incubated for 30 min prior to applying rhinovirus isolates. Uninfected tissues treated with ColdZyme mouth spray alone (UI/CZ) served as negative controls in all experiments. Positive controls were mock-treated rhinovirus-infected tissues (rhinovirus).

 Absolute quantification of viral loads from 3 dpi apical supernatants and 7 dpi basolateral subnatants of

differently treated cells revealed significant protection from infection by applying the ColdZyme mouth spray (Fig. 4 apical (left panel) and basolateral (right panel)). Viral loads in differentially treated tissues were determined by quantitative real-time RT-PCR using a rhinovirus standard. Apically, rhinovirus-infected cultures illustrated a high productive infection with RV35 (mean ∼$1 \times 10^7$ copies/ml) and RV48 (mean ∼$3.5 \times 10^7$ copies/ml) on 3 dpi, which was significantly reduced by a one-time application of ColdZyme mouth spray for both strains. Basolateral release of viruses was only marginally detectable until 5 dpi, while 7 dpi, viral copies were detected for RV35 (∼$1.5 \times 10^4$ copies/ml) and RV48, where copy numbers varied from ∼$1 \times 10^5$ to ∼$3.5 \times 10^7$ copies/ml. In summary, we found that a single application of ColdZyme mouth spray blocked infection of HAE cultures with two strains of non-enveloped rhinovirus by significantly reducing virus release, both apically and basolaterally.

**ColdZyme mouth spray maintains epithelial integrity upon rhinovirus infection of NHBE cultures.** To monitor whether ColdZyme mouth spray protects respiratory tissues from potential rhinovirus-mediated destruction, analysis of TEER (Fig. 5), an indicator for tissue integrity, was performed 3 dpi. TEER values of infected cultures were significantly lower compared with controls or ColdZyme-treated/infected cultures independent of the strain used. While in ColdZyme-treated UI as well as infected tissue cultures TEER values ranged from 900 to 1500 $\Omega/cm^2$, this range dropped to 700 to 1000 $\Omega/cm^2$ in RV35- and RV48-infected cells. Tissue integrity was greatly rescued in infection experiments 3 dpi, if ColdZyme was applied before rhinovirus exposure.

**Disappearance of virus from pseudostratified epithelia upon pre-treatment with ColdZyme mouth spray.** In accordance with viral load analyses and TEER, image analyses upon infection revealed that ColdZyme mouth spray protects the epithelium from interaction with the

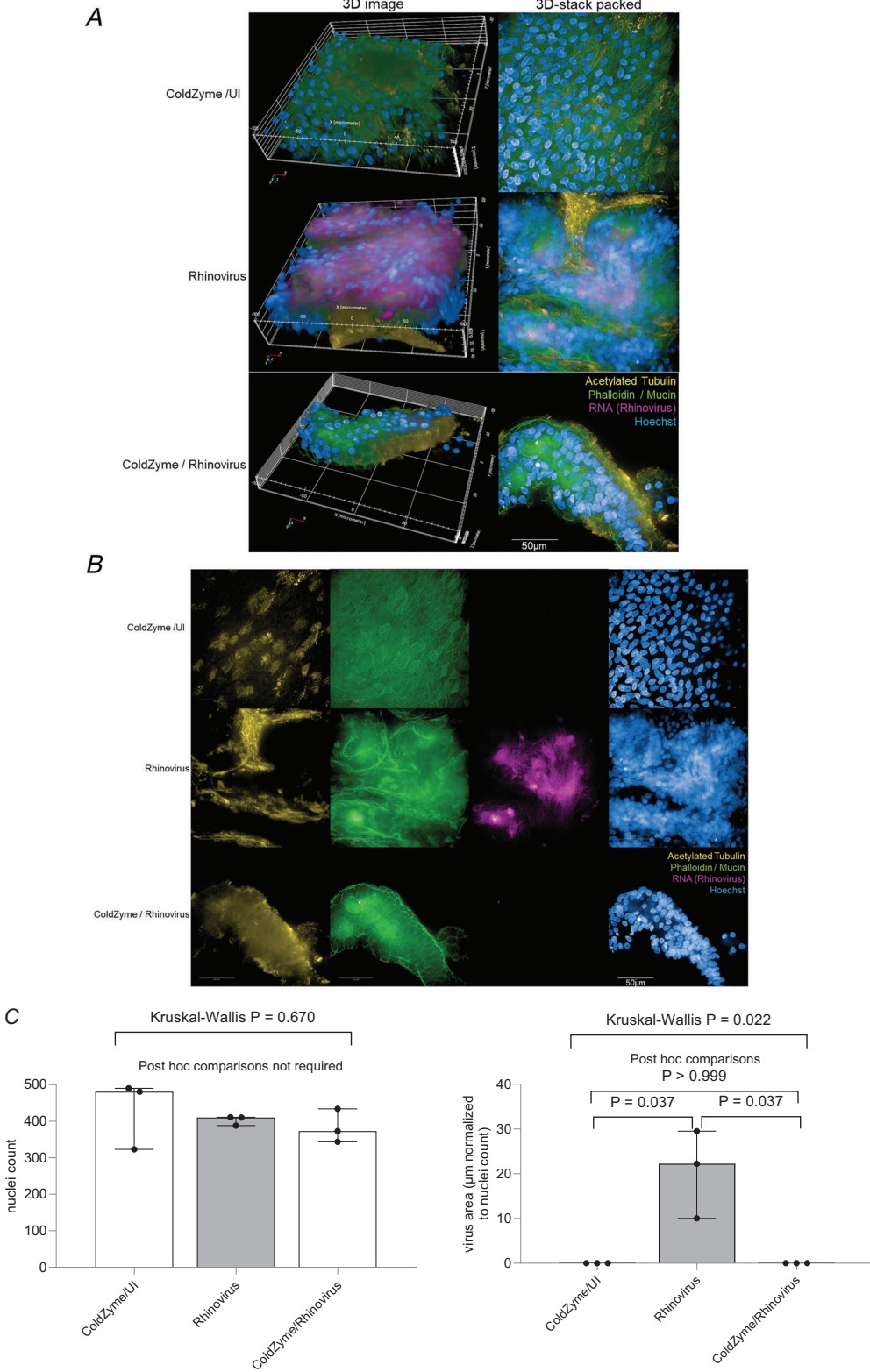

**Figure 1. Visualisation of virus binding (rhinovirus RNA, pink) in ColdZyme-treated or untreated rhinovirus-infected 3D pseudostratified epithelia (*n* = 3 per condition)**

3D image (*A*) and 3D stack packed (*B*), single-colour overviews are shown for ColdZyme uninfected (UI), (panel 1), rhinovirus-infected (panel 2) and ColdZyme-pre-treated and rhinovirus-infected (panel 3). Images were generated

using the Operetta CLS HCS and the 63 × WATER objective. Cells were stained for nuclei using Hoechst (blue), rhinovirus using an antibody recognising viral RNA (pink), actin and mucin using a combination of phalloidin and MUC5AC (green) and cilia using acetylated tubulin (yellow). High virus deposition and nuclear fragmentation were evident in rhinoviurs-infected cultures (middle panels), while no virus and low nuclear destruction were detected in UI (panel 1) and ColdZyme/rhinovirus (panel 3) cultures. Scale bars represent 50 μm. Three independent experiments were performed (from distinct areas of the culture) per condition, each including 300–500 cells with virus area calculated per condition using the Harmony 4.8 software (*C*), nuclei count (*C* left panel) was not different between conditions (overall Kruskal–Wallis test *P* = 0.670 is shown on the figure) so no further *post hoc* comparison was performed. For the normalised virus data (i.e. right panel: virus area:nuclei) Kruskal–Wallis test *P* = 0.022, so results for individual Bonferroni-corrected Mann–Whitney U *post hoc* comparisons between conditions are displayed in the figure. [Colour figure can be viewed at wileyonlinelibrary.com]

viruses. In Fig. 1*A*, 3D analyses of mock-treated, control (upper panel) and rhinovirus-infected (middle panel) cells and ColdZyme-treated, rhinovirus-infected (lower panel) cultures are depicted. In contrast to infected cultures, where the virus signal was detectable over the whole area in mucus layers (Fig. 1*A*, middle panel, yellow–pink overlay), this was not the case for ColdZyme/UI and ColdZyme/rhinovirus-infected cultures, where only signals for mucus and F-actin (yellow) were apparent (Fig. 1*A*, upper and lower panels). A 3D image (left), 3D-stack packed (middle) and input image (right) are illustrated (Fig. 1*A*, upper panels). When switching off signals for phalloidin/MUC5AC and acetylated tubulin and just leaving virus and nucleus signals, the protection by ColdZyme pre-treatment was even more clear (Fig. 1*B*). Only in rhinovirus-infected tissues a clear virus signal was detectable, while in ColdZyme/UI- and ColdZyme/rhinovirus-infected cultures, only a faint background fluorescence was visible (Fig. 1*B*, lower panels). These analyses are in accordance with the viral load data and TEER analyses, illustrating greater exposure and infection of pseudostratified epithelial layers by RV35 and RV48 3 dpi, while prophylactic treatment with ColdZyme mouth spray rescued the tissue from interaction with this non-enveloped common cold virus.

## Discussion

The main findings from this study are: (i) ColdZyme was able to reduce the duration for which URTI symptoms persisted in endurance athletes; (ii) the ColdZyme treatment group reported fewer missed training days as a consequence of URTI episodes; (iii) mechanistically, these findings were supported by lower rhinovirus viral load in swabs obtained from the ColdZyme group compared with the placebo group; and (iv) these findings are further supported by our *in vitro* physiological human airway model, showing protection against rhinovirus release from infected cells, and better maintained epithelial integrity. Together, these data provide robust evidence on the physiological mechanisms that explain the primary findings (i.e. protection against viral release from infected cells, and better maintained airway epithelial cell integrity).

We illustrate here, in a full randomised double-blind placebo-controlled trial that ColdZyme mouth spray used in accordance with the manufacturer's instructions reduced self-reported URTI episode duration (mean reduction of ∼5 days: ∼42%; median reduction ∼2 days: 21% in episode duration). This was also associated with a reduction in lost training days (mean reduction in days lost ∼2.4 days: 60%; median reduction ∼2 days: 67%), which may have implications for athlete performance (Keaney et al., 2019; Pyne et al., 2005; Raysmith & Drew, 2016; Reid et al., 2004). The differences observed between placebo and ColdZyme are even more pronounced in the group with swab-confirmed pathogen detection (see Tables 4 and 5, and Fig. 6). The participants who did not return swabs still recorded episodes that met the Jackson questionnaire criteria, and by this method they can be counted as URTI episodes. However, even assuming they were truly from an infective cause, without the swabs it is not possible to identify which pathogen(s) may be responsible for these episodes. Therefore, it is unclear whether the infection was caused by multiple pathogens or, alternatively, by a single pathogen. It is possible that different pathogens respond differently to ColdZyme, as we have observed here, *in vivo*, for example, *H. influenzae* load in swab samples was less prone to ColdZyme reduction than rhinovirus, which may help to further explain this difference. This highlights the importance of determining the pathogens responsible for the URTI. Of note, all of the compliant episodes with swabs available for analysis (*n* = 50) were positive for rhinoviruses, meaning these data provide a good representation of the magnitude of the effect of ColdZyme treatment on rhinovirus infection (which is the most common cause of a cold/URTI in general). We detected rhinovirus in 100% of cases where swabs were returned, higher than observed in previous studies (e.g. Hanstock et al., 2016; Spence et al., 2007). It is also worth noting that we detected rhinovirus in 14 out of 35 healthy/asymptomatic samples, despite the fact that no self-reported URTI symptoms were present at this time. Previous research has reported asymptomatic rhinovirus infection (e.g. Granados et al., 2015, in a university student population), albeit at a lower rate (i.e. 8% *vs*. 40% seen here). Both of these findings (100% positive detection rate in symptomatic, and 40%

positive in asymptomatic swabs) are likely due to the fact that we used broad rhinovirus targets to ensure full strain coverage (as an illustration of this, if we had used a narrower panel, e.g. only targeted to Vi99990017 assay, we would have detected positives in 38/50 of symptomatic cases – 76%). This demonstrates the importance of having broad strain coverage to ensure all potential pathogens are included in the screening, thus avoiding false negatives (e.g. Kustin et al 2019). It also highlights the importance of collecting symptom data (as we have done with the Jackson questionnaire) as the clinically relevant end-point, with swabs used in a confirmatory way. In addition, pre-

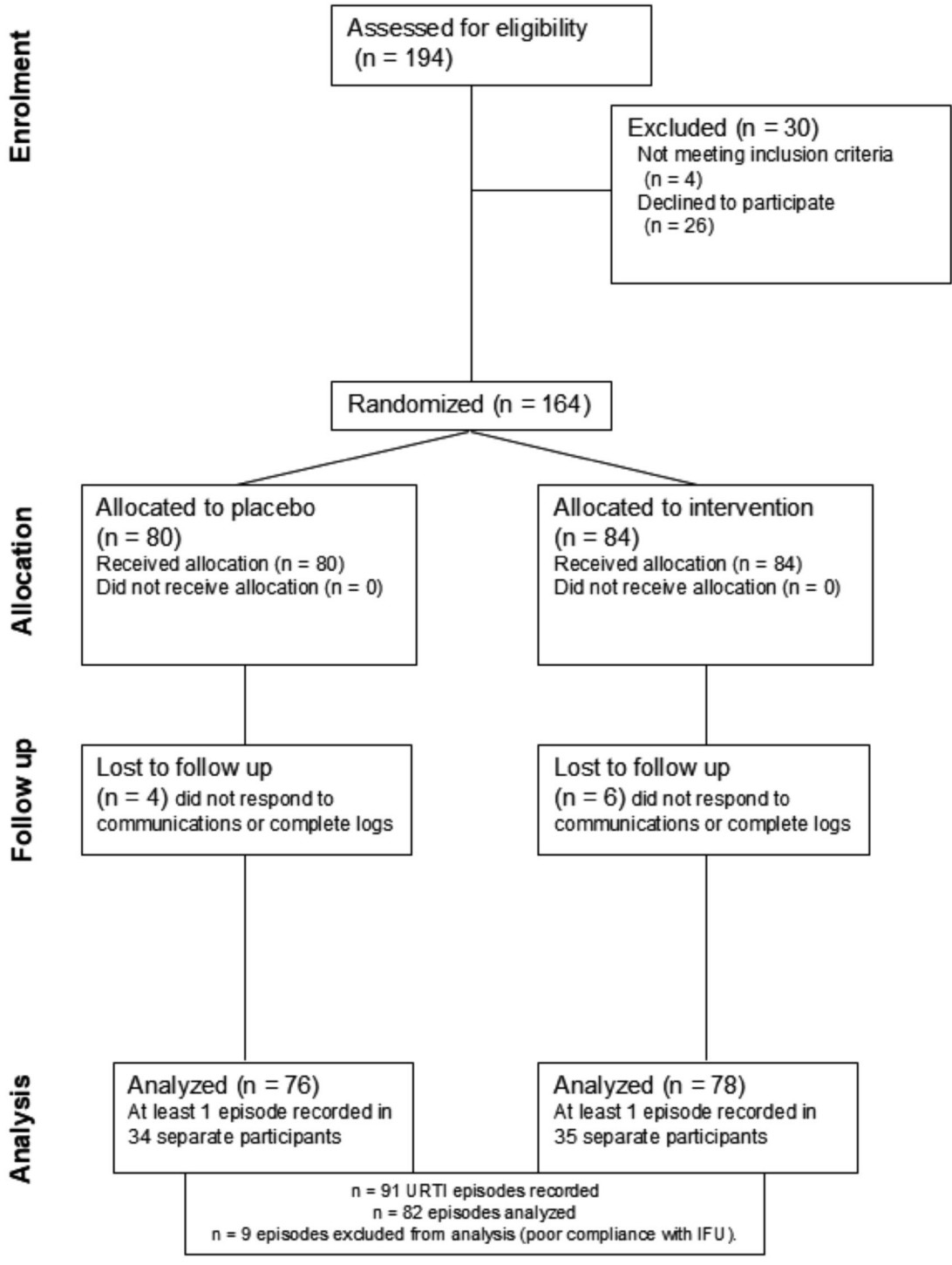

**Figure 2. Consort diagram showing participant recruitment and flow through the study.**

**Table 4. Upper respiratory tract infection (URTI) episodes with swab-confirmed pathogen detection (*n* = 50)**

| | | Placebo | ColdZyme | *P* |
|---|---|---|---|---|
| **URTI episode duration (days)** | | 11.8 ± 10.9 | 5.7 ± 2.4 | 0.003 |
| Mean ± SD | | 7.0 (5.3–13.5) | 5.0 (3.8–7.3) | |
| Median (interquartile range) | | | | |
| **Jackson score (whole episode)** | | 66.2 ± 56.9 | 31.4 ± 25.1 | <0.001 |
| Mean ± SD | | 46.5 (26.0–99.0) | 21.0 (13.8–44.3) | |
| Median (interquartile range) | | | | |
| **Jackson score (mean per day)** | | 6.0 ± 2.3 | 5.0 ± 2.1 | 0.052 |
| Mean ± SD | | | | |
| **Jackson score (episode peak)** | | 10.6 ± 3.9 | 8.1 ± 4.2 | 0.009 |
| Mean ± SD | | 10.0 (7.0–14.0) | 7.0 (5.0–10.3) | |
| Median (interquartile range) | | | | |
| **Jackson individual symptoms** | | | | |
| Sneezing | Mean ± SD | 7.5 ± 5.0 | 4.2 ± 3.4 | 0.004 |
| | Median (interquartile range) | 6.0 (3.0–10.8) | 4.0 (2.0–5.3) | |
| Headache | Mean ± SD | 3.6 ± 4.7 | 2.8 ± 4.0 | 0.172 |
| | Median (interquartile range) | 2.5 (0.0–5.0) | 1.0 (0.0–4.0) | |
| Malaise | Mean ± SD | 9.6 ± 11.0 | 4.0 ± 5.0 | 0.013 |
| | Median (interquartile range) | 7.0 (1.3–13.0) | 2.5 (0.0–5.5) | |
| Nasal discharge | Mean ± SD | 12.5 ± 8.2 | 7.1 ± 3.8 | 0.004 |
| | Median (interquartile range) | 10.0 (5.8–17.8) | 6.0 (4.0–9.3) | |
| Nasal obstruction | Mean ± SD | 11.5 ± 12.3 | 4.8 ± 4.9 | 0.007 |
| | Median (interquartile range) | 7.0 (3.0–11.5) | 3.5 (0.0–9.0) | |
| Sore throat | Mean ± SD | 7.6 ± 6.4 | 4.3 ± 3.7 | 0.016 |
| | Median (interquartile range) | 7.0 (2.0–11.5) | 4.0 (1.0–6.0) | |
| Chilliness | Mean ± SD | 2.5 ± 3.8 | 0.7 ± 1.3 | 0.062 |
| | Median (interquartile range) | 0.0 (0.0–5.3) | 0.0 (0.0–1.3) | |
| Cough | Mean ± SD | 11.3 ± 16.0 | 3.5 ± 5.0 | 0.005 |
| | Median (interquartile range) | 4.5 (2.0–18.0) | 1.5 (0.0–5.0) | |

Statistical tests: independent *t* test on Jackson score (mean per day), or transformed (normalised) data for all other variables except chilliness and cough (could not be normalised), which were compared with Mann–Whitney U tests.

vious athlete studies have used a single swab day, but we tested across 7 days, which reduced the likelihood of missing a positive due to daily fluctuations or false negatives. Our results also suggest that an appropriate questionnaire, applied in the correct way, with the proper application of the scoring criteria, can successfully identify URTI episodes in this population (while swabs for pathogen screening and viral load determination remain important to provide insight into mechanisms for interventions such as this one). There is a wide range of URTI episode durations reported (see Fig. 6), but this is quite normal for studies monitoring naturally acquired URTI (e.g. Cunniffe et al., 2011; Davison et al., 2021; Jones et al., 2014; Valtonen et al, 2019). It is interesting, however, that

the range appears narrower in the placebo group. For example (from the *n* = 82 episodes) there are two outliers in the placebo group and no outliers in the ColdZyme group; and (from the *n* = 50 swab-confirmed episodes) there are four outliers in the placebo group and one in the ColdZyme group. In both cases, the removal of the outliers does not alter the overall outcome that episode duration is shorter in ColdZyme *vs.* placebo groups (*P* = 0.030 and *P* = 0.016, respectively). We also suggest that it would be inappropriate to remove the outliers in this case, and that the lower number seen in the ColdZyme group is not a chance occurrence, and is unlikely to be 'random noise' or error, but is rather a result of the ColdZyme treatment *per se*. That is, the fact that there were fewer

**Table 5. Training days affected by upper respiratory tract infection episodes**

Overall (*n* = 82)

| | Placebo | ColdZyme | *P* |
|---|---|---|---|
| **Training days affected (total)** Mean ± SD Median (interquartile range) | 7.1 ± 9.4 4.5 (2.8–8.3) | 3.5 ± 3.2 3.0 (1.0–6.0) | 0.024 |
| **Training days missed** Mean ± SD Median (interquartile range) | 4.0 ± 4.5 3.0 (0.0–6.0) | 1.6 ± 2.1 1.0 (0.0–3.0) | 0.003 |
| **Training reduced (days)** Mean ± SD Median (interquartile range) | 3.2 ± 5.7 1.5 (0.0–3.0) | 1.8 ± 2.1 1.0 (0.0–3.0) | 0.323 |

Only episodes with pathogen-confirmed swabs returned (*n* = 50)

| | Placebo | ColdZyme | *P* |
|---|---|---|---|
| **Training days affected (total)** Mean ± SD Median (interquartile range) | 8.5 ± 10.6 5.0 (3.0–8.8) | 3.0 ± 3.2 3.0 (0.0–5.0) | 0.003 |
| **Training days missed** Mean ± SD Median (interquartile range) | 4.5 ± 4.9 3.0 (0.5–6.8) | 1.6 ± 2.2 0.0 (0.0–3.0) | 0.004 |
| **Training reduced (days)** Mean ± SD Median (interquartile range) | 4.0 ± 6.4 2.0 (0.3–4.5) | 1.4 ± 1.6 0.5 (0.0–3.0) | 0.042 |

Statistical tests: independent *t* test on transformed (normalised) data for training days missed; Mann–Whitney U tests for training days affected and training days reduced.

outliers (from longer episodes) in the ColdZyme group is likely a reflection of the effectiveness of the treatment.

In addition to the lower episode duration and total Jackson score (i.e. global symptoms score) with

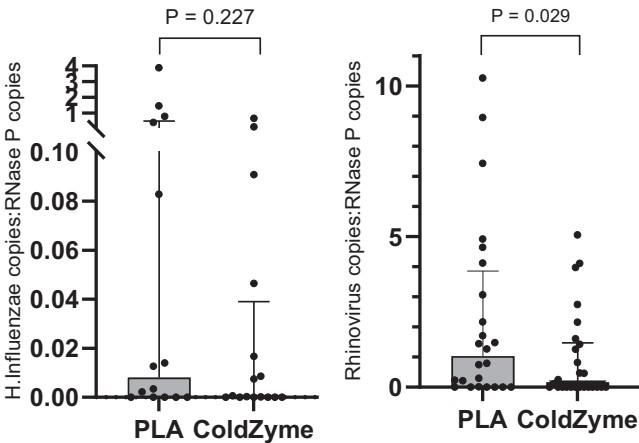

**Figure 3. Absolute quantification for *H. influenzae* (left panel; *n* = 30) and rhinovirus (right panel; *n* = 50)**
Bars show median and interquartile range. *Significant difference between placebo and ColdZyme for rhinovirus (*P* = 0.029). Statistical tests: rhinovirus, independent *t* test on transformed (normalised) data; *H. influenzae*, Mann–Whitney U test.

ColdZyme, there was also a lower peak score in ColdZyme *vs.* placebo (Table 4). This parameter shows the highest single-day Jackson score reported during the episode (i.e. it reflects the 'worst' day experienced, in terms of symptoms, during a given episode), which indicates lower symptom severity ratings with ColdZyme *vs.* placebo within the group of *n* = 50 swab-confirmed episodes. The findings of lower tissue destruction (via TEER) in the *in vitro* airway model provide support for this, as tissue damage and associated inflammation are factors that contribute to URTI symptoms *in vivo* (Blaas & Fuchs, 2016; Warner et al., 2019). It is interesting that the reduction of TEER values with rhinovirus infection that we observed (Fig. 5) were not as prominent as those observed when tissues were infected with SARS-CoV-2 or influenza, which are typically associated with more severe symptoms, and greater damage to epithelial cells (Posch et al., 2021; Zaderer et al., 2022), which further supports this association.

There was no difference between groups on URTI incidence. Furthermore, of the 78 completing participants in the ColdZyme group, 41 reported using the product preventatively (i.e. when they suspected increased risk) at selected times, in addition to using it as treatment (i.e. at the first sign of symptoms). Incidence rates were similar

in this sub-group (i.e. 41% total incidence, $0.51 \pm 0.68$ episodes per person), which was not significantly different from the placebo group ($P = 0.362$). This is likely related to the fact that with 'real-life' application *in vivo*, it is not possible for any protective ('barrier') effects to be evenly and effectively distributed over all relevant surfaces, such as the epithelial cells in the oropharynx and nasopharynx. Furthermore, these effects are unlikely to be sustained 24 h a day as the product will be continually 'washed' off via swallowing, saliva production or normal cellular actions. As a result, repeated 'replenishment' is needed, and windows of opportunity for viral entry can never be completely eliminated. Based on the evidence in this study, it would seem there is limited benefit in preventing infection altogether, rather, effects are seen via reducing local propagation in the URT, reducing the spread of infection to neighbouring cells, and reducing overall viral load compared with the untreated condition. There is also support for the idea that substances may act via anti-pathogenic mechanisms, locally within the oropharynx region, for example, with the reduction in URTI duration that has been reported with the use of zinc lozenges (Hemilä, 2017). However, the zinc lozenge interventions require a daily zinc intake of up to 75 mg, well in excess of the recommended daily levels and even beyond advised upper intake limits, which may present safety concerns. Nevertheless, for the purposes of comparison with the zinc lozenge studies, a meta-analysis by Hemilä (2017) reported a 33% reduction in episode duration, whereas the present findings show a greater reduction (42%) in episode duration with ColdZyme (note, these calculations are based on the relative scale method described by Hemilä (2017), in which the mean episode duration is transformed relative to the mean placebo group duration, which is assigned the value of 100%).

To summarise, the present results suggest that ColdZyme reduces viral load in the local URT region, which subsequently allows more rapid elimination via natural immune defences, and remission of the illness and associated symptoms sooner than would otherwise be the case without treatment. In the present study, there are clear protective effects seen *in vitro*, in our human airway model (with rhinovirus), alongside previous work with other URTI-causing pathogens (e.g. SARS-COV-2, Posch et al., 2021; Zaderer et al., 2022), which does suggest a plausible mechanism to reduce infection from occurring in the first place. However, this might require more frequent, or strategically timed, applications or a method to prolong the persistence of such barrier properties *in vivo*. Alternatively, unlike the *in vitro* model, which has clear control over viral strains, timing of infection and dose of virus (i.e. MOI *in vitro*), in the real-world setting, there are likely large variations between different episodes recorded in each case, which may confound this particular outcome. Direct viral challenge models (e.g. Prather et al., 2015) would be required to determine the effectiveness of ColdZyme in reducing infection incidence under conditions of controlled exposure.

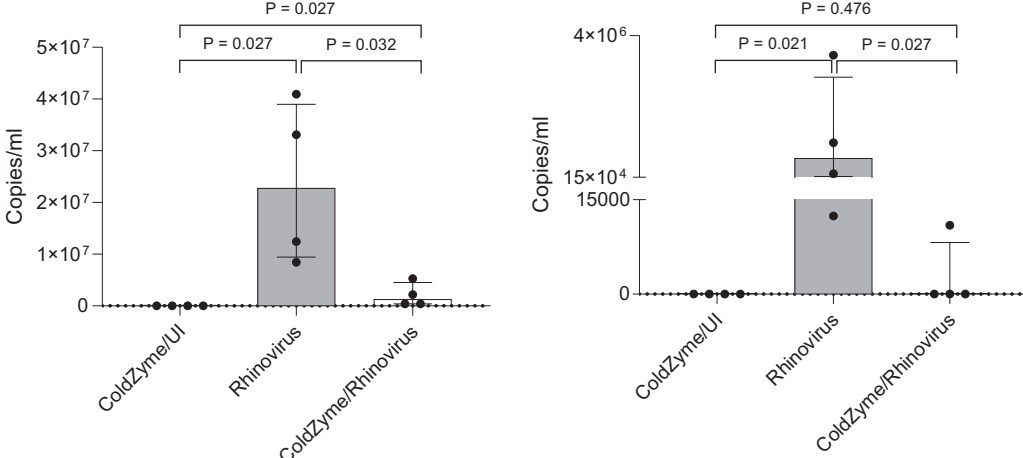

**Figure 4. ColdZyme protects from rhinovirus infection (*n* = 4 per condition)**
Pseudostratified epithelia were infected by apical addition of RV35 and RV48 (MOI 0.01) with (ColdZyme/rhinovirus) or without (rhinovirus) ColdZyme pre-treatment and incubated for up to 7 days post-infection (dpi). ColdZyme/UI: uninfected samples treated with ColdZyme only. Copy numbers of apically (left panel) collected supernatants on 3 dpi and basolateral (right panel) supernatants on 7 dpi. Experiments were repeated four times and RT-PCR analyses were performed in duplicate (with replicate values then averaged). The figure shows median and interquartile range. Kruskal–Wallis overall between the three conditions $P < 0.001$ and with Bonferroni-corrected *post hoc* Mann–Whitney U comparisons between each condition shown in figures.

## Limitations

Self-report methods and illness questionnaires are subjective, which presents the possibility of athletes reporting symptoms/illness in the absence of a true infection. Some studies have reported an increased occurrence of allergy-type symptoms that are often mistaken by athletes as URTI, although this tends to be more common in spring, when responses to environmental allergens such as pollen are more common (Robson-Ansley et al., 2012). The current study was conducted in the winter months when URTI incidence is known to be at a peak. In addition, the validated Jackson questionnaire and scoring criteria were used, which helped to protect against false positive episode counts. Finally, swabs were collected to screen for known URTI-causing pathogens, with at least one known URTI-causing pathogen detected in at least one swab for all that were returned ($n = 50$ episodes).

As noted above, in the real-world setting with naturally occurring infection, there is no control over which pathogens people are exposed to, nor the timing or 'doses' presented. In addition to this, the pathogen screening panel is designed to be as broad as possible, in order to target the maximal range of pathogens, variants and strains (with the aim of determining the infection cause). This means that it is not possible to account for different pathogens and variants, which may have

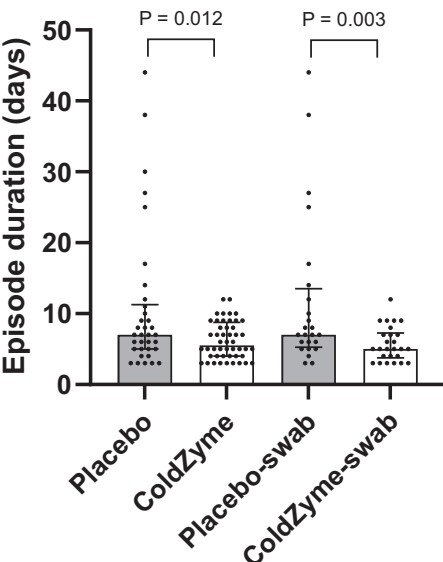

**Figure 6. Upper respiratory tract infection episode duration of placebo *vs*. ColdZyme and swab-confirmed placebo (placebo–swab) *vs*. swab-confirmed ColdZyme (ColdZyme–swab) analysis groups**
Whole group ($n = 82$ episodes), and swab-confirmed ($n = 50$). Bars show median and interquartile range. Statistical tests: independent *t* tests on transformed (normalised) data. Note: whole group placebo group contained two outliers, ColdZyme group zero outliers, if outliers are removed placebo *vs*. ColdZyme comparison $P = 0.030$. Swab-confirmed placebo group contained four outliers, ColdZyme group one outlier, if outliers are removed placebo *vs*. ColdZyme comparison $P = 0.016$. See the Discussion section for further consideration of this.

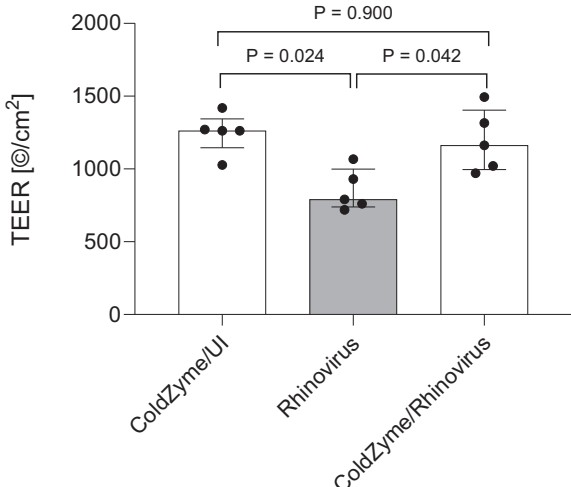

**Figure 5. TEER measurement of epithelial integrity during rhinovirus infection ($n = 5$ per condition)**
Cells were infected by apical addition of RV35 and RV48 (MOI 0.01) with (ColdZyme/rhinovirus) or without (rhinovirus) ColdZyme pre-treatment and incubated. TEER was measured on 3 days post-infection using an EVOM volt-ohm-meter. Experiments are from five independent pseudostratified epithelia measurements performed in triplicate (with replicate values then averaged). Bars show median and interquartile range. Kruskal–Wallis test $P = 0.024$ so results for individual Bonferroni-corrected Mann–Whitney U *post hoc* comparisons between conditions are displayed in the figure.

differing infectivity profiles or cause different symptoms or severity ratings. Future research may benefit from more focused strain-level pathogen analysis, which would allow additional sub-group comparisons to be made between groups infected with the same strains. However, given that the present study was conducted with a relatively large sample, and randomisation was balanced at each geographical location, and within each season/time of year, it is expected that similar pathogens were in circulation and encountered equally between groups.

*In vitro* models also have some limitations due to missing immune components (e.g. immune cells). However, the combination of both *in vivo* and *in vitro* models, as used in the present study, is effective to demonstrate clinically relevant effects and the underpinning mechanisms.

## Conclusions

The results of this study show that ColdZyme is effective at reducing URTI episode duration, but not incidence. The results also show that ColdZyme is effective at reducing symptom ratings and missed training days in endurance

athletes, which are likely explained by reduced viral load. These findings align with our previous open-label study (Davison et al., 2021). However, the present study was a full double-blind placebo-controlled trial, which further strengthens the robustness of our results. This suggests that ColdZyme can prevent local viral propagation, infectivity and spread of the URTI. As a result, epithelial cells suffer less damage and inflammation, symptom severity ratings are lowered, episodes are shorter-lived and the athletes are fit to return to training sooner. These findings are further strengthened by the novel *in vitro* data we present, showing protection against rhinovirus release from infected human epithelial cells, and better maintained epithelial integrity in the human airway model of rhinovirus infection.

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

## Additional information

### Data availability statement

Data are available from the following public data repository: https://data.kent.ac.uk/id/eprint/548.

### Competing interests

The *in vivo* study was funded by Enzymatica AB, Sweden, who are the manufacturers of ColdZyme. The *in vitro* study was partly funded by Enzymatica AB. Both studies were independently, investigator-initiated by the researchers (G.D. *in vivo*; D.W. *in vitro*), and the funders had no influence on analysis, data validation or interpretation.

### Author contributions

G.D. and D.W.: conception of projects; acquisition, analysis and interpretation of data; drafting and revising manuscript; final approval of manuscript; agreement to be accountable for all aspects of the work. M.S., C.C., D.D., G.M., P.S., C.W., W.P., C.C., K.F. and C.M.: acquisition and analysis of data; revising manuscript; final approval of manuscript; agreement to be accountable for all aspects of the work.

### Funding

Authors were supported by the Austrian Science Fund (FWF; P 34 070-B to W.P.), the MUI (MUI-DK02 CONNECT to D.W.) and Enzymatica AB (Sweden) to G.D., and D.W. Austrian Science Fund (FWF): Wilfried Posch, P34070-B. The *in vivo* study was funded by Enzymatica AB, Sweden, who are the manufacturers of ColdZyme. The *in vitro* study was partly funded by Enzymatica AB. Both studies were independently, investigator-initiated by the researchers (G.D. *in vivo*; D.W. *in vitro*), and the funders had no influence on analysis, data validation or interpretation.

### Keywords

common cold, exercise, human airway model, illness, rhinovirus

## Supporting information

Additional supporting information can be found online in the Supporting Information section at the end of the HTML view of the article. Supporting information files available:

**Peer Review History**

