## [Peer Review History · The Journal of Physiology]

ColdZyme® reduces viral load and upper respiratory tract infection duration and protects airway epithelia from infection with human rhinoviruses.

Glen Davison, Marlene Schoeman, Corinna Chidley, Deborah K Dulson, Paul Schweighofer, Christina Witting, Wilfried Posch, Guilherme Matta, Claudia Consoli, Kyle Farley, Conor McCullough, and Doris Wilflingseder
DOI: 10.1113/JP288136

Corresponding author(s): Glen Davison (g.davison@kent.ac.uk)

The following individual(s) involved in review of this submission have agreed to reveal their identity: Graeme Iain Lancaster (Referee #2); Sean Williams (Referee #3)

Review Timeline:

Submission Date:	18-Nov-2024
Editorial Decision:	16-Dec-2024
Revision Received:	24-Jan-2025
Editorial Decision:	31-Jan-2025
Revision Received:	05-Feb-2025
Accepted:	07-Feb-2025

Senior Editor: Harold Schultz

Reviewing Editor: Mike Stemberidge

Transaction Report:

Dear Dr Davison,

Re: JP-RP-2024-288136 "ColdZyme® reduces viral load and upper respiratory tract infection duration and protects airway epithelia from infection with human rhinoviruses." by Glen Davison, Marlene Schoemann, Corinna Chidley, Deborah K Dulson, Paul Schweighofer, Christina Witting, Wilfried Posch, Guilherme Matta, Claudia Consoli, Kyle Farley, Conor McCullough, and Doris Wilflingseder

Thank you for submitting your manuscript to The Journal of Physiology. It has been assessed by a Reviewing Editor and by 3 expert referees and we are pleased to tell you that it is potentially acceptable for publication following satisfactory major revision.

REVISION CHECKLIST:

We look forward to receiving your revised submission.

Yours sincerely,

Harold Schultz
Senior Editor
The Journal of Physiology

REQUIRED ITEMS

- Author photo and profile. First or joint first authors are asked to provide a short biography (no more than 100 words for one author or 150 words in total for joint first authors) and a portrait photograph. These should be uploaded and clearly labelled together in a Word document with the revised version of the manuscript. See Information for Authors for further details.

- You must start the Methods section with a paragraph headed Ethical Approval. If experiments were conducted on humans, confirmation that informed consent was obtained, preferably in writing, that the studies conformed to the standards set by the latest revision of the Declaration of Helsinki and that the procedures were approved by a properly constituted ethics committee, which should be named, must be included in the article file. If the research study was registered (clause 35 of the Declaration of Helsinki), the registration database should be indicated, otherwise the lack of registration should be noted as an exception (e.g. The study conformed to the standards set by the Declaration of Helsinki, except for registration in a database). For further information see: <https://physoc.onlinelibrary.wiley.com/hub/human-experiments>.

- Your manuscript must include a complete Additional Information section, including competing interests; funding; author contributions and acknowledgements.

- Please upload separate high-quality figure files via the submission form.

- Papers must comply with the Statistics Policy: https://jp.msubmit.net/cgi-bin/main.plex?form_type=display_requirements#statistics.

In summary:

- If n {less than or equal to} 30, all data points must be plotted in the figure in a way that reveals their range and distribution. A bar graph with data points overlaid, a box and whisker plot or a violin plot (preferably with data points included) are acceptable formats.

- If $n > 30$, then the entire raw dataset must be made available either as supporting information, or hosted on a not-for-profit repository, e.g. FigShare, with access details provided in the manuscript.

- 'n' clearly defined (e.g. x cells from y slices in z animals) in the Methods. Authors should be mindful of pseudoreplication.

- All relevant 'n' values must be clearly stated in the main text, figures and tables.

- The most appropriate summary statistic (e.g. mean or median and standard deviation) must be used. Standard Error of the

Mean (SEM) alone is not permitted.

- Exact p values must be stated. Authors must not use 'greater than' or 'less than'. Exact p values must be stated to three significant figures even when 'no statistical significance' is claimed.

- Please include an Abstract Figure file, as well as the Figure Legend text within the main article file. The Abstract Figure is a piece of artwork designed to give readers an immediate understanding of the research and should summarise the main conclusions. If possible, the image should be easily 'readable' from left to right or top to bottom. It should show the physiological relevance of the manuscript so readers can assess the importance and content of its findings. Abstract Figures should not merely recapitulate other figures in the manuscript. Please try to keep the diagram as simple as possible and without superfluous information that may distract from the main conclusion(s). Abstract Figures must be provided by authors no later than the revised manuscript stage and should be uploaded as a separate file during online submission labelled as File Type 'Abstract Figure'. Please also ensure that you include the figure legend in the main article file. All Abstract Figures should be created using BioRender. Authors should use The Journal's premium BioRender account to export high-resolution images. Details on how to use and access the premium account are included as part of this email.

EDITOR COMMENTS

Reviewing Editor:

Thank you for submitting to J Physiol. As you can see, two experts have reviewed your manuscript. While they see merit in the study, they both raise significant issues that require addressing. Reviewer 1 questions a number of limitations that are not acknowledged and need addressing. Reviewer two highlights the questionable use of one-tailed statistics, a view that is also shared by our statistics editor. Please pay particular attention to these points.

Please also see 'Required Items' above.

Senior Editor:

Comments for Authors to ensure the paper complies with the Statistics Policy (Required):
Please include actual p values in figures. Do not use symbols or NS. Please indicate in the respective figure legends, the type of data summary [mean (SD) or interquartile range] used in each figure. It is stated in some but not all figure legends.

Comments to the Author:

Thank you for submission of your research article to the Journal of Physiology for consideration. The article has been reviewed by experts in the field and found to require a major revision to address methodological and statistical concerns before a final assessment can be given. This is not a guarantee of final acceptance. Please address all comments from the external referees and reviewing editor, which may require additional experiments, as well as addressing the list of requirements or publication in the journal including the statistical requirements. Please adhere to the Rigor & Reproducibility requirements for the Journal as outlined in the link below.

<https://physoc.onlinelibrary.wiley.com/pb-assets/hub-assets/physoc/documents/TJP-Rigour-and-Reproducibility-Requirements-1724673661727.pdf>

REFeree COMMENTS

Referee #1:

General

The authors investigated whether ColdZyme mouth spray reduces viral load and RTI symptoms in N154 athletes. Their novel findings show that the mouth spray reduced URTI symptom duration and lost training days and reduced viral load for

rhinovirus in-vivo and in an in-vitro model. This is an important study given the burden of respiratory infections and limited preventative and therapeutic treatments. The approaches taken appear appropriate in terms of design, power etc and the authors should be applauded for performing the partner in-vitro work. My specific comments relate to clarity, controls, interpretation and limitations.

Specific

L99: Suggest you firstly emphasize that the prominent risk factors for respiratory infection in athletes are largely the same as non-athletes (autumn-winter, travel, life stress etc.) then you can mention the apparent clustering around heavy training and competition. I think that improves your reach and shows balance.

L131: It should be made clearer here and in the abstract when exactly the spray should be/was used, as either a preventative (L134 "creating a temporary barrier") or to reduce symptom severity and duration during an URTI episode (as a therapeutic). This would help interpretation and practical application. For example, zinc supplementation on a daily basis is not recommended for preventing the common cold but zinc lozenges may shorten a common cold if taken during an episode.

L184: The demographic and training status of the participants would help.

L199: It nice to see efforts were made to collect URTI symptom information using the Jackson questionnaire and that reminders were sent to participants. Did you have an adherence threshold whereby non-compliant participants were removed, say if they completed < 80% of questionnaires?

L206: The study took place from 2022. How do you account for the various other viral causes of URTI e.g., covid-19, RSV, adenovirus etc.? Important limitation for the discussion.

L231: This information describing the mouth spray protocol must be clear in the abstract/highlights. Did you monitor the doses/exposure to the spray as you say L234 "optionally use sprays..."? A limitation.

L246: Excellent that you include swabs in this way. Ack. The limitation raised earlier about limiting to rhinovirus, few studies have attempted to do this.

L253: Please can you make clearer if your Jackson score threshold for a common cold of 14 is for a total score across 3 or more days? Or was the daily score threshold 14 which is too conservative (e.g., 7 of 8 as moderate symptoms or 4 severe and 2 mild on that day!)? That sounds far too high so my guess is the former.

L272: PCR, was quantification provided only for influenza and rhinovirus? Good to expand why other common cold causing targets were not considered or edit accordingly. Also important to raise (discussion) the issue of commensal presence for the targets you report. Only rhinovirus is mentioned in the abstract, contrasting with L272.

L374: The lack of difference in URTI incidence in the experimental and placebo groups is an important finding. Better if this is highlighted in the abstract results. Table 2 shows a trend to greater number of episodes in the ColdZyme group (P 0.107). Likely its important how the participant chooses to use the spray.

L397: Although rhinovirus is highlighted earlier it seems other respiratory targets were assessed.

L560: Nice summary but I'd prefer to see here and in the abstract that there was no influence of ColdZyme on URTI incidence. Stating this important finding does not reduce the impact of your paper - although, it seems, you probably think it

does.

L568: A gap is the dose/exposure to the spray, I don't think you assessed this. How many sprays were they given and what did they use?

Another question is did you provide an exit interview to establish the % of participants who correctly guessed what treatment they received?

L575: Are you suggesting that some of the URTI symptoms were due to non-infectious cause (allergy, other)? L578: true episodes is too strong. Its great you used the Jackson as the original study involved giving cold viruses and assessing symptoms but it doesn't confirm true infection.

L583: please clarify if you are talking about anti-viral vs bacterial. Unclear.

L587: 100% rhinovirus in swabs returned? Some discussion is required that the pathogen analysis provided any/suitable discrimination of infectious aetiology (commensal). L592-5 please see last sentence. If 100% positive pathology the Jackson alone selected your sick vs non-sick.

L623: Would it be easier/cheaper to just take zinc lozenges as atherapeutic? Id be interested in this comparison. I actually think comparing ColdZyme vs zinc lozenges (assuming comparable effects/pathways) is important for those who seek a therapeutic for the common cold. Some mention in the discussion is required. Given the commercial interest Id think this comparison is important.

L640-50: Is some of this an admission that the pathogen analysis didn't provide the level of discrimination you required (100% positive)? I may have misread or misinterpreted, apologies. If I have done so others may misinterpret too. This para would not be needed if you could sensitively/accurately determine those URTI symptom episodes that were truly infectious. L652- this topic continues in the next paragraph (red flag). The adv or disadv of your pathogen analysis needs to be more clearly presented. Arguably a high percentage of those scoring 14 or more over 3 days (by Jackson) would, in the winter, be expected to translate into a high % of positives for respiratory targets (like rhinovirus or sars-cov etc). Other limitations are probably more important given that symptoms are the target - control over compliance, accounting for exposure to the spray (dose, timing etc), exit interview, others deserve mention.

L668: No effect on incidence deserves mention first.

I hope you agree that these comments and suggestions help to improve the paper.

Referee #2:

In the manuscript by Davison et al, a randomized control trial is conducted and demonstrates that a mouth spray (Coldzyme) is able to reduce the burden of URTI in an athlete population. This is a well written and interesting piece of work. The RCT appears well conducted, it is appropriately powered, the experiments and participants were blinded to the intervention and during analysis, an appropriate placebo was used. The work has several strengths, in particular the use of nasal swabs to

confirm infection status and the type of infection present; this complements the use of questionnaires to determine self-reported URTI. The breadth of information collected by the authors is also a strength of the work. In addition, the in vitro studies provide a nice additional layer of mechanistic confirmation regarding the efficacy of the coldzyme spray. I outline my specific comments below.

(1) A major concern is the use of 1-tailed tests to compare between groups (essentially all the important comparisons for all measures used in the study). I am not a statistician, but the use of a 1-tailed test concerns me. The authors justify the use of the 1-tailed because they had a specific directionality regarding the hypothesis (that coldzyme would be beneficial). In my view, that coldzyme may have been detrimental is a feasible outcome and one that could not be tested using a one tailed test. The use of 1-tailed tests is quite uncommon and the concern here is that are the authors using this to gain more statistical power? Was this study pre-registered and was it made clear at that stage that a 1-tailed test would be used? In my view a far stronger argument should be presented by the authors. Furthermore, in my view, because the conclusions of this work are largely dependent on the outputs from performing 1-tailed tests, an independent statistician should be consulted to confirm the validity of using a 1-tailed test in this scenario.

(2) In figure 2, the significant effects appear to be driven by a small number of 'extreme' values. These values seem intuitively unusually long to me - i.e. having a URTI for 5-7 weeks? Can the authors comment on this?

(3) In Figure 4, the data has not been analyzed correctly. 8 samples are shown on the figure, but the figure legend states that this is from 4 independent experiments, and that PCR was performed in duplicate. These duplicate measures should not be treated as biological replicates, but as technical replicates, and hence the true 'n' of this experiment is 4.

(4) The same issue described for figure 4 is also applicable to figure 5. 15 dots are shown from an experiment performed on 5 independent occasions, with measures from each independent experiment being done in triplicate. The 'n' for this experiment is 5, not 15. The analysis for these experiments needs to be redone.

(5) I found the images in figure 6A rather difficult to differentiate between, it's not clear what signal is different. 6B is far clearer, but these are simply representative images from one culture, some group quantification would be beneficial.

(6) While the statistical concerns above must be addressed, this work is well conducted and provides evidence on the effectiveness of the Coldzyme spray. However, in my view this work does not advance physiology knowledge, it merely establishes the potential efficacy of a compound in a RCT. As such, this work seems a rather odd fit for this journal. Furthermore, the same authors in a study published in 2021 came to essentially the same conclusions - albeit the current study was performed to a higher standard (e.g. double blinded, placebo controlled RCT). Accordingly, the findings of the current work lack novelty.

Referee #3:

Comments for Author (Required):

The authors' use of one-sided tests warrants careful reconsideration. While they justify this choice by focusing on beneficial effects, it seems to me that the potential for harm (e.g., ColdZyme increasing the frequency of infections, prolonging URTI duration, or worsening symptoms) cannot be excluded. One-sided tests are only appropriate when there is strong prior evidence to rule out adverse effects, yet the authors provide no discussion of such evidence. Furthermore, their prospective registration lacks a statistical analysis plan, making it unclear whether the use of one-sided tests was pre-specified.

To ensure transparency and scientific rigour, the authors should either provide a robust justification for their use of one-sided tests based on prior studies, or reanalyse their data using two-sided tests to account for both potential benefits and harms. Given the results, most findings are likely to remain statistically significant even with two-sided tests.

I also echo Reviewer 2's concerns regarding pseudoreplication in Figures 4 and 5. The duplicate PCR measurements should be averaged or otherwise summarised to reflect a single value per biological replicate. Alternatively, a linear mixed model (with random effects for biological replicates) could be used to address the hierarchical structure of the data.

Finally, the use of a Kruskal-Wallis test with post hoc Mann-Whitney U comparisons in Figure 4 is not mentioned or justified in the Methods section. The authors should provide a rationale for these analyses and ensure proper control of the type I error rate, such as using Bonferroni corrections.

END OF COMMENTS

Dear Editor,

We would like to thank the editors and reviewers for their careful and thoughtful evaluation of our initial submission and providing us with specific comments and suggestions to improve the overall quality of our manuscript. The following responses have been prepared to address the editor and reviewers' comments in a point-by-point fashion. For clarity the reviewers' comments are copied in bold text and yellow highlight, with our responses beneath each comment in standard black text. Changes in the manuscript are shown via the MS Word track changes function (a clean copy is also provided). We believe that this has significantly improved our paper and hope that this is now to your satisfaction.

Yours sincerely

Glen Davison & Doris Wilflingseder (corresponding authors).

EDITOR COMMENTS

Reviewing Editor:

Thank you for submitting to J Physiol. As you can see, two experts have reviewed your manuscript. While they see merit in the study, they both raise significant issues that require addressing. Reviewer 1 questions a number of limitations that are not acknowledged and need addressing. Reviewer two highlights the questionable use of one-tailed statistics, a view that is also shared by our statistics editor. Please pay particular attention to these points.

Please also see 'Required Items' above.

Thank you for your careful consideration and constructive comments. We have addressed all of the required items as requested.

Senior Editor:

Comments for Authors to ensure the paper complies with the Statistics Policy (Required):

Please include actual p values in figures. Do not use symbols or NS. Please indicate in the respective figure legends, the type of data summary [mean (SD) or interquartile range] used in each figure. It is stated in some but not all figure legends.

Thank you. We have now added actual P values in all cases.

Comments to the Author:

Thank you for submission of your research article to the Journal of Physiology for consideration. The article has been reviewed by experts in the field and found to require a major revision to address methodological and statistical concerns before a final assessment can be given. This is not a guarantee of final acceptance. Please address all comments from the external referees and reviewing editor, which may require additional experiments, as well as addressing the list of requirements or publication in the journal including the statistical requirements. Please adhere to the Rigor & Reproducibility

requirements for the Journal as outlined in the link below.

<https://physoc.onlinelibrary.wiley.com/pb-assets/hub-assets/physoc/documents/TJP-Rigour-and-Reproducibility-Requirements-1724673661727.pdf>

Many thanks. We have added all relevant references and catalogue numbers etc where appropriate.

REFEREE COMMENTS

Referee #1:

General

The authors investigated whether ColdZyme mouth spray reduces viral load and RTI symptoms in N154 athletes. Their novel findings show that the mouth spray reduced URTI symptom duration and lost training days and reduced viral load for rhinovirus in-vivo and in an in-vitro model. This is an important study given the burden of respiratory infections and limited preventative and therapeutic treatments. The approaches taken appear appropriate in terms of design, power etc and the authors should be applauded for performing the partner in-vitro work. My specific comments relate to clarity, controls, interpretation and limitations.

Thank you for the detailed evaluation and constructive feedback. We believe that our revisions have significantly improved the paper and hope that this is now to your satisfaction.

Specific

L99: Suggest you firstly emphasize that the prominent risk factors for respiratory infection in athletes are largely the same as non-athletes (autumn-winter, travel, life stress etc.) then you can mention the apparent clustering around heavy training and competition. I think that improves your reach and shows balance.

This has now been added. Please see lines 114-115 of the revised submission (tracked changes version).

L131: It should be made clearer here and in the abstract when exactly the spray should be/was used, as either a preventative (L134 "creating a temporary barrier") or to reduce symptom severity and duration during an URTI episode (as a therapeutic). This would help interpretation and practical application. For example, zinc supplementation on a daily basis is not recommended for preventing the common cold but zinc lozenges may shorten a common cold if taken during an episode.

Thank you. We have added some further explanation (lines 153-156 of the revision, tracked changes submission), which hopefully clarifies this. The proposed mechanism is the same for both preventative or treatment use. That is, even for use after first symptoms the mechanism is still via forming a barrier, that will be present to limit viral spread when infected cells release them, and thus reduce spread and infection to neighbouring cells. This might explain why preventative use did not stop infection altogether (which I suspect is also the case for the zinc lozenge studies etc

too)... i.e. it was not possible for any barrier to be universally present on all possible infection surfaces at all times, to an extent that can prevent infection from happening at all, but rather it can slow down viral propagation and spread, allowing a more speedy elimination and resolution of the episode (see also lines 729-739 in the revised manuscript).

L184: The demographic and training status of the participants would help.

Demographic information is included in the results (lines 406-411), and is self-reported by participants. Training status was not determined by objective physiological testing (e.g. VO₂max etc) so we could not classify participants by this type of metric.

L199: It nice to see efforts were made to collect URTI symptom information using the Jackson questionnaire and that reminders were sent to participants. Did you have an adherence threshold whereby non-compliant participants were removed, say if they completed < 80% of questionnaires?

All included participants had log completion compliance over 90%. The data on compliance with product use instructions was of most importance to us (which was also recorded in the logs). Based on the results from our previous study, we excluded those for whom compliance with the product IFU was poor (which occurred in 9 cases- i.e. below 5 uses per day, which equates to ~80% compliance with IFU guidelines)- see lines 417-419 for further detail on this.

L206: The study took place from 2022. How do you account for the various other viral causes of URTI e.g., covid-19, RSV, adenovirus etc.? Important limitation for the discussion.

Thank you for this important point. This is accounted for in several ways. 1. The patterns of pathogens detected were similar across seasons. 2. The number of participants were balanced between groups (placebo and ColdZyme) within season, and within testing sites (geographical location). All participants that returned swabs were positive for RV, although the 31 cases where swabs were not returned is a limitation (this has been included in the discussion). The full range of pathogens identified is provided in the results, see lines 444-447 of the revised submission, (with per subject data available in the full deposited data set, so this will be openly available after publication).

We also believe that we have addressed this potential limitation in the 2nd paragraph of the limitations section however (please see lines 774- 776 of the revised manuscript).

L231: This information describing the mouth spray protocol must be clear in the abstract/highlights.

Apologies for the confusion caused with the way this was phrased. We have added further clarification to this text (please see lines 260-266). This also relates to/follows on from the comment above about the mechanism of action, which we hope the above explanation further clarifies (i.e. the mechanisms is always as barrier, acting to deactivate viruses and prevent spread), which we hope helps provide further clarification in relation to this point also. We have elaborated further on this and it is also discussed further in the discussion (lines 642-644). Unfortunately, there is not sufficient allowance in the word count to provide this level of detail in the abstract. We also feel that the highlights section is not an appropriate location for this level of

methods information and detail. We think this is OK.

Did you monitor the doses/exposure to the spray as you say L234 "optionally use sprays..."? A limitation.

Yes, we had to do this to monitor compliance, since those with poor compliance were excluded. This was recorded daily by participants in their logs (please see further info in the methods section). In accordance with our previous study we excluded those for whom compliance was not over 80% (which equates to using the spray 5 or more times per day). We have now added further explanation (lines 417-419), which we hope further clarifies this point.

L246: Excellent that you include swabs in this way. Ack. The limitation raised earlier about limiting to rhinovirus, few studies have attempted to do this.

Thankyou. Hopefully the response above also addresses this in relation to limitation statement?

L253: Please can you make clearer if your Jackson score threshold for a common cold of 14 is for a total score across 3 or more days? Or was the daily score threshold 14 which is too conservative (e.g., 7 of 8 as moderate symptoms or 4 severe and 2 mild on that day!)? That sounds far too high so my guess is the former.

Thankyou. It is the daily score. This is the standard method applied for this scoring system (see Martineau et al., 2015 for example). Please also note, however, that this is one of the reasons for the other 2 criteria, where criterion 3 sometimes comes into play to ensure you don't exclude episodes that are likely a true URTI. (i.e. "...or iii) a total Jackson symptom score <14 + subjective impression of having a cold + nasal discharge for at least 3 days").

L272: PCR, was quantification provided only for influenza and rhinovirus? Good to expand why other common cold causing targets were not considered or edit accordingly. Only rhinovirus is mentioned in the abstract, contrasting with L272.

Apologies for any confusion here. To clarify, a comprehensive screening panel was performed, as detailed in this section (i.e. using the TrueMark™ Respiratory Panel 2.0 array). This was done initially to determine presence or absence of known URTI-causing pathogens. Following this initial screening, absolute quantification was performed on the two most abundant pathogens (Haemophilus influenzae and Rhinovirus targets) with specific targeted qPCR assays performed in triplicate. Further explanation and elaboration is provided later (i.e. lines 464-469). In total 24 different pathogens were detected across samples (Adenovirus, H.influenzae, Rhinovirus, Epstein-Barr virus, Human herpesvirus 6, Influenza, Parainfluenza, Respiratory syncytial virus, Metapneumovirus, Measles, Mumps, Coronaviruses, MERS CoV, SARS CoV, SARS-CoV2 (Covid-19), Enterovirus, Bordetella, B.pertussis, K.pneumoniae, M.catarrhalis, M.pneumoniae, S.aureus, S.pneumoniae, P.jirovecii), many of which were co-infections.

However, only H.influenzae (30 episodes: 14 Placebo, 16 ColdZyme) and Rhinovirus (50 episodes: 24 Placebo, 26 ColdZyme) were detected with sufficient frequency in each group to warrant meaningful statistical comparisons on relative viral and bacterial loads (see Table 4 and Figure 3), and so it was not warranted (or feasible) to perform the fully quantitative determination for the other 20+ targets.

Also important to raise (discussion) the issue of commensal presence for the targets you report.

We hope that the additional discussion added at lines 669-674 and 678-681 answers/addresses this sufficiently.

L374: The lack of difference in URTI incidence in the experimental and placebo groups is an important finding. Better if this is highlighted in the abstract results. Table 2 shows a trend to greater number of episodes in the ColdZyme group (P 0.107). Likely its important how the participant chooses to use the spray.

We have highlighted the lack of difference in incidence between groups to the abstract as requested (please see line 83). It is correct that compliance with instructions for proper use of the product can affect this, as shown in our previous study, which we hope the responses above have further clarified. Regarding the $P = 0.107$ value, we would prefer (in line with best practice on statistics and reporting) not to report this as a “trend” or make any inferences beyond this. We would prefer to stick with reporting the data, and precise P values, as we have done here (which we believe to be in line with the Journal’s statistics policy) which show no difference. We hope you will agree that the current presentation in the abstract is appropriately focussed in relation to the primary outcome and hypotheses, bearing in mind the strict 250 word limit for the abstract.

L397: Although rhinovirus is highlighted earlier it seems other respiratory targets were assessed.

That is correct. Hopefully the responses above, and associated amendments, clarify and address this comment too?

L560: Nice summary but I prefer to see here and in the abstract that there was no influence of ColdZyme on URTI incidence. Stating this important finding does not reduce the impact of your paper - although, it seems, you probably think it does.

Thank you, we agree and did not think this reduced the impact. However, as briefly mentioned in the response above, it was down to space in the abstract and the need to prioritise other results, which we believe are the key findings in relation to the primary outcome. We have found a little space to add this extra comparison, however, which we hope is to your satisfaction.

L568: A gap is the dose/exposure to the spray, I don't think you assessed this. How many sprays were they given and what did they use?

Sorry if this was not clear. Please see also responses above- we assessed this via the daily logs (see compliance comments, and lines 417-419 or the revised submission) because we observed in our previous study that those with low compliance did not experience the same level of benefit as those with high compliance. As such, we excluded 9 cases where compliance was low. These cases were not sufficient to allow separate statistical comparison of this as an additional sub-group (i.e. good vs poor IFU compliance) in the way we did in our previous study.

Another question is did you provide an exit interview to establish the % of participants who correctly guessed what treatment they received?

Thank you for highlighting this important consideration. This was not included in our initial exit survey, but we sent an additional “post-study survey” to all completing participants after completion of the study. We have added an additional section to the results (lines 493-501 of the revised submission) where these data are summarised.

L575: Are you suggesting that some of the URTI symptoms were due to non-infectious cause (allergy, other?)?

This is always a possibility, although we are not saying this was necessarily the case here. The intention with this paragraph was to highlight this as a possibility. We have generally seen good agreement between episodes determined according to the Jackson criteria, and confirmation of pathogens in swabs, so because the Jackson criteria were also met in these 31 cases, we believe it is highly likely that the we would have also detected pathogens in the majority, if not all, of these cases if their swabs were returned to us. However, we are unable to confirm this without the swabs. There is, therefore, a possibility that some of the cases with missing swabs would have actually been negative, and hence not an infectious cause that can be influenced by the ColdZyme treatment. This is strengthened as a possibility by the more notable differences that emerge in the group that did return their swabs. However, it is not possible to fully confirm this in the absence of the missing swabs. We hope that this consideration comes across in the rest of this paragraph, and the discussions therein (e.g. lines 659-663 and lines 679-699).

L578: true episodes is too strong. Its great you used the Jackson as the original study involved giving cold viruses and assessing symptoms but it doesn't confirm true infection.

We hope the additional now added help to better clarify this now. This also relates to the above point/response. Since we have followed accepted procedures for quantifying and scoring the Jackson questionnaire, we do believe that it is appropriate to report these outcomes accordingly (i.e. as episodes or not), which was the intention here. However, for the avoidance of any doubt we have amended our wording in this section (see lines 654-656). As noted previously, it is highly likely (given the criteria etc) that the cause was infective. However, we also believe that the next sentence within that paragraph is the more important point, which is that without the swabs we cannot know this for certain, and most importantly, even assuming an infective cause, we do not know which pathogen(s) they are infected with (e.g. “whether the infection was caused by multiple pathogens or, alternatively, by a single pathogen” since it is “possible that different pathogens respond differently to ColdZyme”).

L583: please clarify if you are talking about anti-viral vs bacterial. Unclear.

That you, this is a very useful point. Although that was the initial hypothesis, based on these data, it is perhaps more accurate to acknowledge that different pathogens may respond differently (linked to the comment and response above) and so we have amended this text accordingly (please see lines 659-662).

L587: 100% rhinovirus in swabs returned? Some discussion is required that the pathogen analysis provided any/suitable discrimination of infectious aetiology (commensal).

Please see further discussions at lines 669-674; and lines 679-681, and 774-776 in the limitations section, which we hope covers the 2nd point.

L592-5 please see last sentence. If 100% positive pathology the Jackson alone selected your sick vs non-sick.

We agree, this does suggest that an appropriate questionnaire, applied in the correct way, with proper application of the scoring criteria, can successfully discriminate “sick vs non-sick”. However we do not believe that this means the swabs are not valuable, or needed, in studies like this one, for several reasons:

1. for objective confirmation of pathogenic cause, and 2. (perhaps most importantly in intervention studies like this), to provide information on potential mechanisms of action (i.e. reduced viral load in this case). We hope this message comes across more clearly now, with the revisions mentioned above.

L623: Would it be easier/cheaper to just take zinc lozenges as a therapeutic? I'd be interested in this comparison. I actually think comparing ColdZyme vs zinc lozenges (assuming comparable effects/pathways) is important for those who seek a therapeutic for the common cold. Some mention in the discussion is required. Given the commercial interest I'd think this comparison is important.

This is a very interesting consideration. There may be some similarities in the mechanisms of action (i.e. ionic zinc/free zinc acting as an ‘antiviral’ that may prevent viral replication, and binding). We agree that some brief comparison may be warranted, and have added a short comparison to the discussion (see lines 729-739). It is worth noting that in the zinc lozenge studies, the required doses of zinc (75 mg or more per day) is many times higher than recommended intake levels (e.g. UK RDA up to 9.5 mg/d, US up to 15 mg/day) and even in excess of tolerable upper intake limits, which I would not want to recommend as an alternative, however.

L640-50: Is some of this an admission that the pathogen analysis didn't provide the level of discrimination you required (100% positive)? I may have misread or misinterpreted, apologies. If I have done so others may misinterpret too. This para would not be needed if you could sensitively/accurately determine those URTI symptom episodes that were truly infectious. L652- this topic continues in the next paragraph (red flag). The adv or disadv of your pathogen analysis needs to be more clearly presented. Arguably a high percentage of those scoring 14 or more over 3 days (by Jackson) would, in the winter, be expected to translate into a high % of positives for respiratory targets (like rhinovirus or sars-cov etc).

Thank you for raising this point and providing an opportunity for us to clarify. That was not how we intended that text to be interpreted. We think it is important, however, not mention the potential limitations with self-report methods used in isolation, especially given the reports from previous studies that have suggested disagreement between positive detection in swabs and outcomes of self-report questionnaires (e.g. Spence et al., 2007; Robson-Ansley et al., 2012; and others). The intention here was to acknowledge this, but to also highlight that this potential limitation is largely mitigated in the current study for a number of reasons: 1. The study was conducted in the Winter months so unlikely to be confounded by the allergy-type symptoms suggested by Robson-Ansley et al. (2012); 2. Application of objective criteria applied to the Jackson questionnaire (as discussed above, and in response to the comment about successfully discriminating “sick from non-sick”. I think this agrees with your point in the last sentence above (hopefully)? So, we would not interpret this as “*the pathogen analysis didn't provide the level of discrimination you required*”, but

rather that it confirmed and validated the use of the Jackson questionnaire to identify URTI episodes. We hope that the additional amendments and discussion points on this (noted above in previous responses and the associated amendments) also help to better clarify and communicate this message now. In relation to the point on advantages and disadvantages of the analysis methods, that was also one of the intentions of the next paragraph, highlighting the challenges faced with “real-world” studies with naturally acquired infections, which we also hope aligns with these points.

Finally, as also noted above, the URTI outcome data (from the Jackson questionnaire) was indeed the primary outcome measure for this study, and our data supports this measure as a valid one for the aim of assessing our primary outcome. However, we do not believe this detracts from the importance of the swabs for several reasons, 1. for objective confirmation of pathogenic cause, and 2. (most importantly) to provide information on potential mechanisms of action in intervention studies like ours (i.e. reduced viral load in this case). With this data, in combination with our novel in vitro data from the human airway model challenged with live virus (Rhinovirus in this case), we can make firmer conclusions about the possible mechanisms of action (i.e. lower viral load; reduced epithelial cell damage), that would not be possible without such measures. So the purpose of their inclusion here, is to provide more robust and rigorous data on the physiological mechanisms that explain the primary findings.

Other limitations are probably more important given that symptoms are the target - control over compliance, accounting for exposure to the spray (dose, timing etc), exit interview, others deserve mention.

We agree, and hope you will agree that these other limitations have all been appropriately addressed and clarified in the responses above and associated revisions.

L668: No effect on incidence deserves mention first.

We think it is important to keep the primary outcome first (in line with the pre-registered trial log), so we have added this as the second point rather than the first.

I hope you agree that these comments and suggestions help to improve the paper.

Thank you once again for the thorough and thoughtful review of our work. We hope that the revisions are to your satisfaction and you agree that the paper is much improved as a result.

Referee #2:

In the manuscript by Davison et al, a randomized control trial is conducted and demonstrates that a mouth spray (Coldzyme) is able to reduce the burden of URTI in an athlete population. This is a well written and interesting piece of work. The RCT appears well conducted, it is appropriately powered, the experiments and participants were blinded to the intervention and during analysis, an appropriate placebo was used. The work has several strengths, in particular the use of nasal swabs to confirm infection status and the type of infection present; this complements the use of questionnaires to determine

self-reported URTI. The breadth of information collected by the authors is also a strength of the work. In addition, the in vitro studies provide a nice additional layer of mechanistic confirmation regarding the efficacy of the coldzyme spray. I outline my specific comments below.

Thank you for your careful evaluation of our work and positive comments and feedback. We believe that our revisions have significantly improved the paper and hope that this is now to your satisfaction.

(1) A major concern is the use of 1-tailed tests to compare between groups (essentially all the important comparisons for all measures used in the study). I am not a statistician, but the use of a 1-tailed test concerns me. The authors justify the use of the 1-tailed because they had a specific directionality regarding the hypothesis (that coldzyme would be beneficial). In my view, that coldzyme may have been detrimental is a feasible outcome and one that could not be tested using a one tailed test. The use of 1-tailed tests is quite uncommon and the concern here is that are the authors using this to gain more statistical power? Was this study pre-registered and was it made clear at that stage that a 1-tailed test would be used?

In my view a far stronger argument should be presented by the authors. Furthermore, in my view, because the conclusions of this work are largely dependent on the outputs from performing 1-tailed tests, an independent statistician should be consulted to confirm the validity of using a 1-tailed test in this scenario.

Thank you for raising this important point. To answer the pre-registration question, yes it was pre-registered (please see methods, and link here to trial registration: <https://doi.org/10.1186/ISRCTN18133939>). Within this prospectively registered trial entry, we did state directional hypotheses as follows:

Null1: There will be no difference between treatment and placebo groups on self-report upper respiratory tract infection (URTI) duration.

Alternate1: Self-report URTI duration will be significantly shorter in the treatment group.

We hope this is sufficient to justify the choice to employ 1-tailed approaches to the statistical analysis, and also reassure reviewers that this was a prospective decision made at the planning stages of the study, before any data collection commenced. This also formed the basis of our power calculations to determine the required sample size. Importantly, we believe that the data used to inform this, from our previous study (2021 study) and others (as cited in the introduction of the paper) provide compelling and robust evidence that this expected directionality is warranted (i.e. and confirm that a negative effects [e.g. increased URTI episode duration; or worsened symptom scores; or increased viral load] etc, was very unlikely). We hope this is sufficient, and in line with the recommendation from the statistics editor/reviewer 3, to provide the requested justification for this approach.

(2) In figure 2, the significant effects appear to be driven by a small number of 'extreme' values.

There were 2 outliers in the original data (44 days and 38 days) in the placebo group but none in the ColdZyme group (and 4 + 1 when looking only at the swab-confirmed group). In both groupings (n = 81 whole group, or n = 50 swab-confirmed group) removal of the outliers does not change the outcome (i.e. significant difference remains, with P remaining below 0.05 in both cases)... please see

additional analysis confirming this now added to figure 2 caption (lines 431-436 of the revised manuscript, tracked changes version). Following on from this point, despite the additional analysis, we believe this should only be used to confirm that the differences persist despite the outliers, but also caution that it would be inappropriate to remove them because they actually represent the true results. That is, the fact that there were no such outliers in the ColdZyme group is likely a reflection of the effectiveness of the treatment (i.e. if that group had also received placebo we would have expected a similar number of episodes of these longer durations, which is normal in other studies of naturally acquired URTI, which is now also highlighted at lines 687-699)- in other words the presence of the longer episodes in the placebo group and absence of this in the ColdZyme group is not a chance occurrence or the result of spurious data, but is rather a result of the effects of the ColdZyme treatment per se.

These values seem intuitively unusually long to me - i.e. having a URTI for 5-7 weeks? Can the authors comment on this?

There were a few episodes that lasted longer than 2 weeks, but we only collected swabs up to day 7, so it is not possible to know what other pathogens they may have presented with at later times during the episode, or as secondary infections. As an example, we have seen reports of the “100 day cough” in circulation, in adults, in the UK for both of these winter periods (caused by pertussis infection- see e.g. for discussion of this infection in adults: <https://pubmed.ncbi.nlm.nih.gov/11360208/>), and this infection (and the associated symptoms) can persist for more than a few weeks, so it is not uncommon or unexpected for a few participants to have reported episodes of this duration.

The longer durations are not uncommon in similar studies (although sometimes this is not overtly visible if they only group average values). Please see further discussion added additional reference to this at lines 687-699 of the revised submission (tracked changes version).

(3) In Figure 4, the data has not been analyzed correctly. 8 samples are shown on the figure, but the figure legend states that this is from 4 independent experiments, and that PCR was performed in duplicate. These duplicate measures should not be treated as biological replicates, but as technical replicates, and hence the true 'n' of this experiment is 4.

Thank you, and apologies for this mistake on our part. We have reanalysed the data in this way and revised Figure 4 accordingly.

(4) The same issue described for figure 4 is also applicable to figure 5. 15 dots are shown from an experiment performed on 5 independent occasions, with measures from each independent experiment being done in triplicate. The 'n' for this experiment is 5, not 15. The analysis for these experiments needs to be redone.

Thank you, and apologies for this mistake on our part. We have reanalysed the data and revised Figure 5 accordingly.

(5) I found the images in figure 6A rather difficult to differentiate between, it's not clear what signal is different. 6B is far clearer, but these are simply representative images from one culture, some group quantification would be

beneficial.

Many thanks for the suggestion. We have now added group qualification graphs to this figure (Figure 6C), and changed the colours within the pictures to make the signals of the single staining even clearer, which we hope achieves this to your satisfaction.

(6) While the statistical concerns above must be addressed, this work is well conducted and provides evidence on the effectiveness of the Coldzyme spray. However, in my view this work does not advance physiology knowledge, it merely establishes the potential efficacy of a compound in a RCT. As such, this work seems a rather odd fit for this journal. Furthermore, the same authors in a study published in 2021 came to essentially the same conclusions - albeit the current study was performed to a higher standard (e.g. double blinded, placebo controlled RCT). Accordingly, the findings of the current work lack novelty.

Thank you. It is true that this study advances the previous one, but we believe there are important additional and novel data and insights that were not previously possible, which together represent a significant advancement of knowledge on this research area, and more broadly. We believe the journal is the perfect fit for this work, given alignment with several areas of the journal's scope, including those related to Human, Exercise, Molecular, and respiratory areas of physiology, and that our findings provide novel insights of relevance to these and other areas. We believe that there are clear physiological implications and consequences of URTIs, not least those in the airway epithelia barrier (e.g. <https://pubmed.ncbi.nlm.nih.gov/37773065/>) so investigations on potential interventions to mitigate this are of value. In addition, we believe that our application of the air-liquid interface and organoid model of the human airways, where we demonstrate protective effects to the epithelial barrier, has real physiological significance. There are notable physiological consequences of damage to this physiological barrier, caused by URT infections, so intervention research is valuable. Importantly, we show alignment between the in vivo results and the human airway in vitro model, which have implications for the validity, and wider applicability of this model to in vivo contexts is an important addition here also (also in support of the 3 Rs principles in physiology research, supporting the use of such models as a legitimate replacement for animal models). It is very common for in vivo and in vitro studies to be done in complete isolation, but we believe that the combination, and alignment, of the two methods in the present study is of practical value to the field. Moreover, often in vivo studies are performed in mice and not in physiologically relevant human 3D models, which we believe is another reason that our study represents an advancement in this field.

Referee #3:

The authors' use of one-sided tests warrants careful reconsideration. While they justify this choice by focusing on beneficial effects, it seems to me that the potential for harm (e.g., ColdZyme increasing the frequency of infections, prolonging URTI duration, or worsening symptoms) cannot be excluded. One-sided tests are only appropriate when there is strong prior evidence to rule out adverse effects, yet the authors provide no discussion of such evidence.

Furthermore, their prospective registration lacks a statistical analysis plan, making it unclear whether the use of one-sided tests was pre-specified.

And

To ensure transparency and scientific rigour, the authors should either provide a robust justification for their use of one-sided tests based on prior studies, or reanalyse their data using two-sided tests to account for both potential benefits and harms. Given the results, most findings are likely to remain statistically significant even with two-sided tests.

Thank you for raising this important point. Although we did not include an analysis plan in the register, we did prospectively register the directional hypothesis (i.e. Null1: There will be no difference between treatment and placebo groups on self-report upper respiratory tract infection (URTI) duration; Alternate1: Self-report URTI duration will be significantly shorter in the treatment group). In addition to this, we employed the same approach in the previous study (Davison et al., 2021), which also formed the basis of our power calculations as reported in register. We hope that you will agree that the previous study (2021 study) and others cited in the manuscript (e.g. Clarsund 2016a; Clarsund 2017a; Clarsund 2017b), as well as the in vitro evidence for reducing viral load for URTI-causing pathogens (e.g. Stefansson et al., 2017; Posch et al., 2021; Zaderer et al., 2022) provide compelling and robust justification that this expected directionality is warranted and also confirms that it was acceptable to exclude the potential for harm as a highly unlikely outcome. We hope this is sufficient to provide the necessary justification for this approach, and also reassure reviewers that this was a prospective decision made at the planning stages of the study, before any data collection commenced.

I also echo Reviewer 2's concerns regarding pseudoreplication in Figures 4 and 5. The duplicate PCR measurements should be averaged or otherwise summarised to reflect a single value per biological replicate. Alternatively, a linear mixed model (with random effects for biological replicates) could be used to address the hierarchical structure of the data.

Thank you, and apologies for this mistake on our part. We have reanalysed the data and amended the figures accordingly.

Finally, the use of a Kruskal-Wallis test with post hoc Mann-Whitney U comparisons in Figure 4 is not mentioned or justified in the Methods section. The authors should provide a rationale for these analyses and ensure proper control of the type I error rate, such as using Bonferroni corrections.

Thank you, we have now added the required details on this to the methods section.

Dear Dr Davison,

Re: JP-RP-2025-288136R1 "ColdZyme® reduces viral load and upper respiratory tract infection duration and protects airway epithelia from infection with human rhinoviruses." by Glen Davison, Marlene Schoemann, Corinna Chidley, Deborah K Dulson, Paul Schweighofer, Christina Witting, Wilfried Posch, Guilherme Matta, Claudia Consoli, Kyle Farley, Conor McCullough, and Doris Wilflingseder

Thank you for submitting your manuscript to The Journal of Physiology. It has been assessed by a Reviewing Editor and by 3 expert referees and we are pleased to tell you that it is acceptable for publication following satisfactory revision.

REVISION CHECKLIST:

We look forward to receiving your revised submission.

Yours sincerely,

Harold Schultz
Senior Editor
The Journal of Physiology

REQUIRED ITEMS

- You must upload original, uncropped western blot/gel images (including controls) if they are not included in the manuscript. This is to confirm that no inappropriate, unethical or misleading image manipulation has occurred. These should be uploaded as 'Supporting information for review process only'. Please label/highlight the original gels so that we can clearly see which sections/lanes have been used in the manuscript figures. For more information, see: <https://physoc.onlinelibrary.wiley.com/hub/journal-policies#imagmanip>.

- Papers must comply with the Statistics Policy: https://jp.msubmit.net/cgi-bin/main.plex?form_type=display_requirements#statistics.

In summary:

- If $n \leq 30$, all data points must be plotted in the figure in a way that reveals their range and distribution. A bar graph with data points overlaid, a box and whisker plot or a violin plot (preferably with data points included) are acceptable formats.

- If $n > 30$, then the entire raw dataset must be made available either as supporting information, or hosted on a not-for-profit repository, e.g. FigShare, with access details provided in the manuscript.

- 'n' clearly defined (e.g. x cells from y slices in z animals) in the Methods. Authors should be mindful of pseudoreplication.

- All relevant 'n' values must be clearly stated in the main text, figures and tables.

- The most appropriate summary statistic (e.g. mean or median and standard deviation) must be used. Standard Error of the Mean (SEM) alone is not permitted.

- Exact p values must be stated. Authors must not use 'greater than' or 'less than'. Exact p values must be stated to three significant figures even when 'no statistical significance' is claimed.

EDITOR COMMENTS

Reviewing Editor:

Thank you for taking the time to revise your manuscript in response to the reviewers' comments. There remains a few issues that require clarification.

Please also see 'Required Items' above.

Senior Editor:

Comments for Authors to ensure the paper complies with the Statistics Policy (Required):

In the data analysis section, please indicate that summary data for parametric analysis was standard deviation (SD). The Journal does not support use of standard error of the mean (SEM). In this section for non parametric analysis, please indicate that summary data are median and relevant interquartile ranges (as used in the figures), or other method used.

In figure and table legends as relevant, please indicate sample sizes and statistical test used for the table/figure.

Comments to the Author:

Thank you for submission of your research article to the Journal of Physiology for consideration. The article has been reviewed by the original referees and found to be potentially acceptable for publication pending further revision to address all of the concerns raised. Please address all comments from the external referees and statistical and reviewing editors as well as addressing the list of requirements or publication in the journal including the statistical requirements below.

In the data analysis section, please indicate that summary data for parametric analysis was standard deviation (SD). The Journal does not support use of standard error of the mean (SEM). In this section for non parametric analysis, please indicate that summary data are median and relevant interquartile ranges (as used in the figures).

In figure and table legends as relevant, please indicate sample sizes and statistical test used for the table/figure.

REFeree COMMENTS

Referee #1:

Thank you to the authors for making the requested changes and for their responses. These are mostly appropriate and improve the manuscript.

There remains one data analysis issue that needs resolving. The authors should be commended for using the Jackson common cold questionnaire but there has been a misunderstanding by a number of authors about the symptom score to determine a common cold. This confusion stems from daily vs total symptom scores over a designated monitoring period. The authors of this JPhysiol submission refer to papers by Martineau to support an apparent symptom score of 14 or more 'per day' to determine a common cold (with answers to yes/no for having a cold and nasal discharge), see Line 278 in revised manuscript. The Martineau papers have misinterpreted the score to determine a common cold in the original Jackson paper. Tables 3 (Mean Total Symptom Score) and 4 in the Jackson 1958 paper (AMA archives of internal medicine, 101(2), 267-278) and text on pp 271 para 2 highlight that the symptom score cut-off (score 14) is the total for a 6-day monitoring period, not a daily score as you have used (see also my initial comment below for example). A total score of 24 in an infected and sick individual, for example (pp 271), would equate to a score each day of 4. In line with this, others have used a daily symptom score to indicate a common cold of 6 or higher (e.g., Gwaltney et al., 1980, Journal of Infectious Diseases, 142(6), 811-815; Cohen et al., 2003, Psychological science, 14(5), 389-395).

'Initial comment' L253: Please can you make clearer if your Jackson score threshold for a common cold of 14 is for a total score across 3 or more days? Or was the daily score threshold 14 which is too conservative (e.g., 7 of 8 as moderate symptoms or 4 severe and 2 mild on that day!)? That sounds far too high so my guess is the former.

'Author Response' Thankyou. It is the daily score. This is the standard method applied for this scoring system (see Martineau et al., 2015 for example). Please also note, however, that this is one of the reasons for the other 2 criteria, where criterion 3 sometimes comes into play to ensure you don't exclude episodes that are likely a true URTI. (i.e. "...or iii) a total Jackson symptom score <14 + subjective impression of having a cold + nasal discharge for at least 3 days").

Referee #2:

I appreciate the work of the authors to fully respond to my comments. Overall, I think they have done an excellent job of addressing all reviewer comments.

I'm pleased that the decision to use a 1 tailed test was pre-registered. However, I'm still concerned about the justification for doing so. While I appreciate the authors have previously shown a beneficial effect of their intervention, I don't really see that this is a strong justification for using a one tailed test. The possibility, even if small, that the spray may have had a detrimental effect is one that surely should have been tested. There are many examples of clinical trials where a supposedly beneficial treatment, that I'm sure had a lot a lot of pre-clinical data supporting its efficacy, ended up having a negative effect. As I indicated initially, I am not a statistician and the decision as to whether the authors approach is justified/conventional for the field, should be made by a statistician with expertise in RCTs.

Referee #3:

I am satisfied with the justification for the use of one-sided tests, particularly the reference to the pre-registered directional hypotheses, the alignment with the previous study (Davison et al., 2021), and the supporting evidence from prior research and in vitro studies. To ensure full transparency, I would recommend adding a brief statement in the text (e.g., alongside the sentence: "All p-values for comparisons between groups are presented as the 1-sided value owing to the directional (i.e. 1-tailed) hypotheses") clarifying that these directional hypotheses were pre-registered.

Thank you for addressing the concern regarding pseudoreplication in Figures 4 and 5. I appreciate the reanalysis of the data and the amendments made to the figures. However, it is not entirely clear from the manuscript or your response which specific approach was taken to address this issue (e.g., averaging duplicate PCR measurements or using a linear mixed model with random effects for biological replicates). I presume it's the former, but could you please clarify the method used within the manuscript so that this is transparent.

Thank you for addressing the concern regarding post hoc Mann-Whitney U comparisons. I would also recommend explicitly stating in the relevant figure captions that the Mann-Whitney U comparisons are Bonferroni-corrected.

END OF COMMENTS

Dear Editor,

We would like to thank the editors and reviewers for their rigorous evaluation of our submission, and for the opportunity to respond. The following responses have been prepared to address the comments in a point-by-point fashion. For clarity the reviewers' comments are copied in bold text and yellow highlight, with our responses beneath each comment in standard black text. Changes in the manuscript are shown via the MS Word track changes function (a clean copy is also provided). We believe that this has significantly improved our paper and hope that this is now to your satisfaction.

Yours sincerely

Glen Davison & Doris Wilflingseder (corresponding authors).

Referee #1:

Thank you to the authors for making the requested changes and for their responses. These are mostly appropriate and improve the manuscript.

There remains one data analysis issue that needs resolving. The authors should be commended for using the Jackson common cold questionnaire but there has been a misunderstanding by a number of authors about the symptom score to determine a common cold. This confusion stems from daily vs total symptom scores over a designated monitoring period. The authors of this JPhysiol submission refer to papers by Martineau to support an apparent symptom score of 14 or more 'per day' to determine a common cold (with answers to yes/no for having a cold and nasal discharge), see Line 278 in revised manuscript. The Martineau papers have misinterpreted the score to determine a common cold in the original Jackson paper. Tables 3 (Mean Total Symptom Score) and 4 in the Jackson 1958 paper (AMA archives of internal medicine, 101(2), 267-278) and text on pp 271 para 2 highlight that the symptom score cut-off (score 14) is the total for a 6-day monitoring period, not a daily score as you have used (see also my initial comment below for example). A total score of 24 in an infected and sick individual, for example (pp 271), would equate to a score each day of 4. In line with this, others have used a daily symptom score to indicate a common cold of 6 or higher (e.g., Gwaltney et al., 1980, Journal of Infectious Diseases, 142(6), 811-815; Cohen et al., 2003, Psychological science, 14(5), 389-395).

Many thanks for pointing this out. We have reviewed all of the other potential episodes to determine whether the application of the daily score threshold at ≥ 6 (rather than 14) would result in the need for any previously excluded reports to be included. In the majority of cases this did not change the determination (usually because the symptoms were reported for 2 days or less, and/or because the total daily scores were 5 or lower). There was one additional episode, however, that was previously excluded when using the daily score of 14 threshold. We have therefore added this case into the results and recalculated all of the relevant values, and rerun the necessary stats on the $n = 82$ cases, instead of the $n = 81$ that we used previously. We have amended all results, figures and tables, as necessary to include

this additional case. There are no notable changes to outcomes. We did not receive any swabs from this individual, however, so the data for the $n = 50$ cases with swabs available for analysis remains unaltered. Please see lines 276-286 of the revised (clean copy) submission for correction to the methods description, and updated data in all relevant tables and figures etc (e.g. lines 81-83 (abstract); 410-411; 415; 419-420; Table 1; Figure 2; Table 2; Table 5; lines 509-512; 642; 683)

Referee #2:

I appreciate the work of the authors to fully respond to my comments. Overall, I think they have done an excellent job of addressing all reviewer comments.

I'm pleased that the decision to use a 1 tailed test was pre-registered. However, I'm still concerned about the justification for doing so. While I appreciate the authors have previously shown a beneficial effect of their intervention, I don't really see that this is a strong justification for using a one tailed test. The possibility, even if small, that the spray may have had a detrimental effect is one that surely should have been tested. There are many examples of clinical trials where a supposedly beneficial treatment, that I'm sure had a lot a lot of pre-clinical data supporting its efficacy, ended up having a negative effect. As I indicated initially, I am not a statistician and the decision as to whether the authors approach is justified/conventional for the field, should be made by a statistician with expertise in RCTs.

Many thanks for the opportunity to respond and hopefully clarify our justification further. Apologies if we have misunderstood, but it was our understanding that the specialist statistics reviewer (reviewer 3) was brought on for exactly this purpose. Since their initial review comments made specific reference to your initial points about this, we assumed they were invited as an expert statistician to specifically focus on this point. If so, we hope that their acceptance of our justification is sufficient to justify this as an acceptable approach. In addition to this, it is also worth noting that the evidence and data that we used to determine a directional hypotheses are more substantial than just pre-clinical data. It is true that there are a lot of pre-clinical, in vitro data etc that support the rationale for our expected directional hypothesis for the in vitro measures. However, in addition to our previous study (Davison et al., 2021), there are also several other in vivo studies in humans that also contribute to this evidence and justification, all showing reductions in subjective symptoms ratings and durations (e.g. Clarsund 2016a; Clarsund 2016b; Clarsund 2017a; Clarsund 2017b; Lindberg et al., 2023). We hope this is now sufficient to provide the necessary justification for this approach.

Referee #3:

I am satisfied with the justification for the use of one-sided tests, particularly the reference to the pre-registered directional hypotheses, the alignment with the previous study (Davison et al., 2021), and the supporting evidence from prior research and in vitro studies. To ensure full transparency, I would

recommend adding a brief statement in the text (e.g., alongside the sentence: "All p-values for comparisons between groups are presented as the 1-sided value owing to the directional (i.e. 1-tailed) hypotheses") clarifying that these directional hypotheses were pre-registered.

Many thanks for this suggestion. We have added this to the revised submission (revised clean copy, lines 367-368).

Thank you for addressing the concern regarding pseudoreplication in Figures 4 and 5. I appreciate the reanalysis of the data and the amendments made to the figures. However, it is not entirely clear from the manuscript or your response which specific approach was taken to address this issue (e.g., averaging duplicate PCR measurements or using a linear mixed model with random effects for biological replicates). I presume it's the former, but could you please clarify the method used within the manuscript so that this is transparent.

Apologies that this was not clear in the response. We can confirm that it was the former approach. We have now added additional explanation to the paper to make this clearer (please see Figure legends: lines 548 and 572).

Thank you for addressing the concern regarding post hoc Mann-Whitney U comparisons. I would also recommend explicitly stating in the relevant figure captions that the Mann-Whitney U comparisons are Bonferroni-corrected.

Many thanks, we have added this to the relevant figure captions (please see relevant captions at lines 550, 574, and 618).

Dear Professor Davison,

Re: JP-RP-2025-288136R2 "ColdZyme® reduces viral load and upper respiratory tract infection duration and protects airway epithelia from infection with human rhinoviruses." by Glen Davison, Marlene Schoeman, Corinna Chidley, Deborah K Dulson, Paul Schweighofer, Christina Witting, Wilfried Posch, Guilherme Matta, Claudia Consoli, Kyle Farley, Conor McCullough, and Doris Wilflingseder

We are pleased to tell you that your paper has been accepted for publication in The Journal of Physiology.

Yours sincerely,

Harold Schultz
Senior Editor
The Journal of Physiology

If you would like to receive our 'Research Roundup', a monthly newsletter highlighting the cutting-edge research published in The Physiological Society's family of journals (The Journal of Physiology, Experimental Physiology, Physiological Reports, The Journal of Nutritional Physiology and The Journal of Precision Medicine: Health and Disease), please click this link, fill in your name and email address and select 'Research Roundup':
<https://www.physoc.org/journals-and-media/membernews>

- You can help your research get the attention it deserves! Check out Wiley's free Promotion Guide for best-practice recommendations for promoting your work at: www.wileyauthors.com/eeo/guide. You can learn more about Wiley Editing Services which offers professional video, design, and writing services to create shareable video abstracts, infographics, conference posters, lay summaries, and research news stories for your research at: www.wileyauthors.com/eeo/promotion.

EDITOR COMMENTS

Reviewing Editor:

Thank you for taking the time to address your manuscript. All outstanding issues have been addressed.

Senior Editor:

The editors wish to thank the authors for these final adjustments to the manuscript. The article is now accepted for publication. Congratulations for an interesting and insightful study. Please consider the Journal of Physiology for your future studies.

REFEREE COMMENTS

Referee #1:

Thank you for making the changes.

Referee #2:

My comments have been addressed.

Referee #3:

I appreciate the revisions made, and I am satisfied that my comments have been addressed.